# LLMBoost: Make Large Language Models Stronger with Boosting

## Abstract

Ensemble learning of LLMs has emerged as a promising alternative to enhance performance, but existing approaches typically treat models as black boxes, combining the inputs or final outputs while overlooking the rich internal representations and interactions across models. In this work, we introduce LLMBoost, a novel ensemble fine-tuning framework that breaks this barrier by explicitly leveraging intermediate states of LLMs. Inspired by the boosting paradigm, LLMBoost incorporates three key innovations. First, a *cross-model attention mechanism* enables successor models to access and fuse hidden states from predecessors, facilitating hierarchical error correction and knowledge transfer. Second, a *chain training paradigm* progressively fine-tunes connected models with an error-suppression objective, ensuring that each model rectifies the mispredictions of its predecessor with minimal additional computation. Third, a *near-parallel inference paradigm* pipelines hidden states across models layer by layer, achieving inference efficiency approaching single-model decoding. We further establish the theoretical foundations of LLMBoost, proving that sequential integration guarantees monotonic improvements under bounded correction assumptions. Extensive experiments on commonsense reasoning and arithmetic reasoning tasks demonstrate that LLMBoost consistently boosts accuracy while reducing inference latency.

## 1 Introduction

Large Language Models (LLMs) have revolutionized natural language processing, building on the Transformer's attention mechanism (Vaswani et al., 2017). Individual models such as GPT-4 (Achiam et al., 2023), Llama-3 (Touvron et al., 2023), and Qwen (Yang et al., 2025) have demonstrated remarkable performance across a wide range of domains. Meanwhile, ensemble learning of LLMs — combining multiple models to yield stronger, more robust performance — is gaining attention (Chen et al., 2025; Lu et al., 2024). However, despite its promise, the exploration of ensembles in the LLM era remains relatively nascent.

Most existing LLM ensemble methods treat each model as a black box and fuse only their final outputs—either at the token level or the full-response level (Chen et al., 2025)—via majority voting (Li et al., 2024a; Yu et al., 2024), weighted aggregation (Li et al., 2024b), or routing strategies (Shnitzer et al., 2023). However, these approaches ignore the rich information embedded in the hidden representations of the LLMs, and more fine-grained interactions—beyond mere output fusion—are needed to fully exploit complementary knowledge across models.

In this paper, we go beyond conventional output-level ensembling and propose a novel fine-tuning framework for LLM ensembles, LLMBoost, which enables fine-grained internal interactions by progressively leveraging the hidden representations within each model in the boosting style. Our main contributions are summarized as follows.

[**Model Architecture**]. We design a residual cross-model attention mechanism in which successor models can adaptively access and fuse the intermediate hidden states of predecessor models, rather than relying solely on its final outputs. The cross-model attention is constructed at the layer level, breaking the black-box barrier and enabling state-driven hierarchical interaction among LLMs. In

addition, progressively targeted fusion is applied to output logits, which allows LLMBOOST to transfer valid knowledge across models effectively.

[**Chain Training Paradigm**]. We introduce a sequential training framework for connected LLMs, where a backbone model is trained with LoRA adapters to preserve general knowledge from pre-training, and each subsequent model is fine-tuned based on the training log of its predecessor. Inspired by the boosting paradigm, the role of a successor model is to rectify the key tokens of its predecessor. Thus, we propose an error-suppression objective, which suppresses the maximal (incorrect) logit produced by the predecessor while enhancing the correct one. By doing so, LLMBOOST significantly reduces the overall computational requirement while ensuring that performance is progressively boosted across the chain of models.

[**Near-Parallel Inference Paradigm**]. We propose a near-parallel streaming inference paradigm to minimize the latency introduced by sequential interactions among LLMs. Instead of waiting for the predecessor to complete full sequence generation, LLMBOOST transmits hidden states at the layer level without delay: the hidden state from the $(l-1)$-th layer of the predecessor is immediately passed to the $l$-th layer of the successor for parallel computation, even before the forward pass is completed. This pipelined design significantly reduces inference latency, achieving performance close to that of a single model while substantially outperforming vanilla inference ensembles.

[**Theoretical Analysis**]. We provide a formal theoretical analysis for LLMBOOST, establishing a principled foundation for its sequential error-correction mechanism. We prove that, under the condition that each new model effectively corrects its predecessor's errors, the ensemble's performance is guaranteed to improve with each model added. This guarantee holds when the contribution of each new model is appropriately weighted, demonstrating a clear theoretical path toward progressively enhancing prediction accuracy through our boosting-style integration. Our analysis is grounded in Mean Squared Error, offering a direct and comprehensive measure of performance gains.

[**Empirical Performance**] . Extensive experiments on commonsense and arithmetic reasoning tasks — including a custom set co-developed with a specific laboratory— confirm our approach's effectiveness. Compared to other ensemble baselines, LLMBOOST consistently enhances performance across tasks in tests: it delivers an overall average improvement of 3.9%, with an even larger 6.6% as the maximum improvement, with 3% improvement observed specifically on Toolchain Dataset(TCD). Moreover, our near-parallel decoding strategy reduces inference latency by 47% compared to conventional sequential ensembles, demonstrating clear advantages in efficiency.

## 2 RELATED WORK

**LLM Ensemble Methods**. Existing LLM ensemble methods primarily focus on output fusion using various strategies. First, inference-time methods enable dynamic integration during decoding, such as token-level probability aggregation (Yao et al., 2024; Yu et al., 2024; Xu et al., 2024b; Li et al., 2024b) and span-level integration (Xu et al., 2024a; Fu et al., 2025). Other methods operate on the final outputs through majority voting (Li et al., 2024a), weighted fusion (Guha et al., 2024), routing-based selection (Ong et al., 2024; Sikeridis et al., 2025), and prompt-based ensembles (He et al., 2024). Second, post-inference methods fuse complete outputs, either via non-cascaded aggregation (Li et al., 2024a; Guha et al., 2024) or cascaded strategies such as difficulty-aware revision (Yue et al., 2023). Third, pre-inference methods focus on task–model matching, including pre-trained routing (Ong et al., 2024; Shnitzer et al., 2023) and non-pretrained routing (Sikeridis et al., 2025; Zhang et al., 2025). While effective, these approaches treat models largely as independent black boxes, overlooking the internal representations and cross-model interactions that could further enhance ensemble performance.

**Boosting Methods.** Boosting (Friedman, 2001; Hastie et al., 2009; Ke et al., 2017) is a classical ensemble paradigm that builds strong learners by sequentially correcting the errors of prior learners. Gradient Boosting minimizes loss by fitting new learners to the residuals of previous outputs (Hastie et al., 2009; Chen & Guestrin, 2016), while AdaBoost reweights samples to emphasize misclassified instances and combines learners via weighted voting (Freund & Schapire, 1997; Li et al., 2008). Recent work has extended boosting to LLMs: Agrawal et al. (2025) applied AdaBoost to train a series of weak LLMs into a stronger model[1], but this approach ensembles over training samples

---

[1]We did not find open-source code, and thus we exclude it from experimental comparisons.

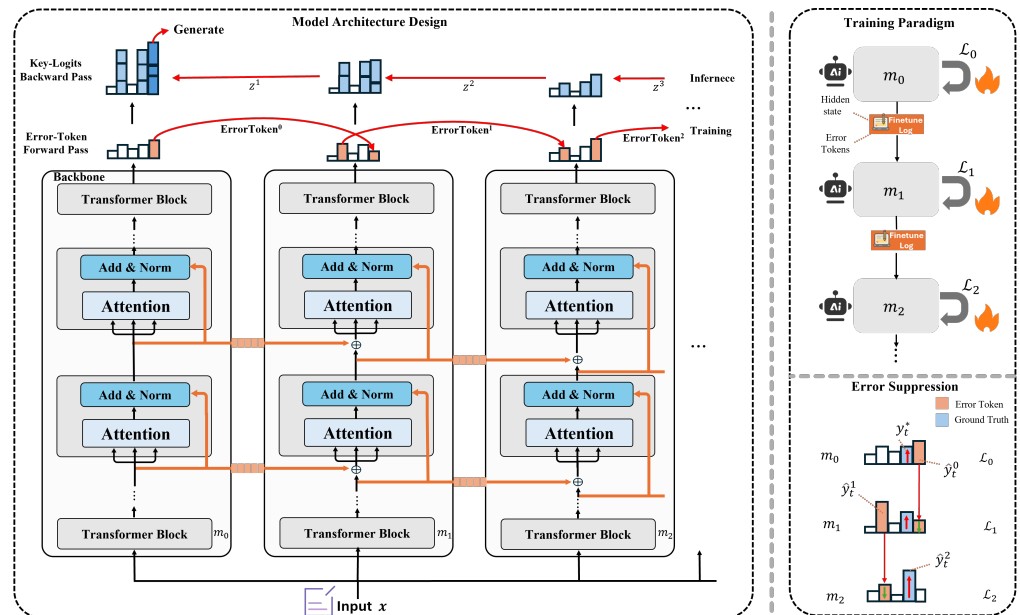

Figure 1: LLMBOOST Framework. The overall framework comprises three key components: (1) LLMBOOST Model Design, (2) Training Paradigm, and (3) Error-Suppression Objective.

while still ignoring internal interactions among LLMs. Zou et al. (2025) introduced copilot adapters to correct errors during model fine-tuning, but the scope is limited to a single auxiliary adapter.

Distinct from existing LLM ensemble works, our work advances the field by ensembling on internal representations. LLMBOOST enables subsequent models to access and fuse predecessors' intermediate hidden states—not just final outputs—for precise error correction, addressing the gap of insufficient intermediate interaction in existing methods while ensuring stable performance gains.

## 3 PROPOSED FRAMEWORK: LLMBOOST

We introduce LLMBOOST, a hierarchical ensemble fine-tuning framework that enables LLMs to perform internal interactions during both training and inference. In the following sections, we elaborate on LLMBOOST from three perspectives: its architecture, training paradigm, and inference paradigm.

**Notations**. As illustrated in Figure 1, LLMBOOST is designed as a sequential ensemble of pretrained LLMs, where each LLM adopts a decoder-only Transformer architecture. Suppose there are $(1 + n)$ LLMs in total. Formally, the ensemble can be represented as $\mathcal{M} = \{m_0(\cdot; \theta_0), m_1(\cdot; \theta_1), \ldots, m_n(\cdot; \theta_n)\}$, where we use $m_i \in \mathcal{M}$ to denote the $i$-th model and $\theta_i$ its trainable parameters for simplicity. In one fine-tuning round $\tau$, we sample an input–output pair $(X_\tau, Y_\tau)$ from a data distribution $D_{\mathcal{X}, \mathcal{Y}}$, where $X_\tau$ is the input sequence and $Y_\tau$ the corresponding target sequence. LLMBOOST generates the output sequence $\widehat{Y}_\tau$ in an autoregressive manner, aiming to approximate the target sequence $Y_\tau$. At the $t$-th decoding step, let $z_t^{(i)}$ denote the logits produced by the $i$-th model, and let $p_t^{(i)} = \mathrm{softmax}(z_t^{(i)})$ be the corresponding probability distribution over the vocabulary. The predicted token of the $i$-th model is then $\hat{y}_t^{(i)} = \arg\max p_t^{(i)}$, while the ground-truth token is denoted by $y_t^*$.

### 3.1 FRAMEWORK ARCHITECTURE

**Cross-Model Attention.** A key innovation of LLMBOOST lies in its fine-grained token-level refinement mechanism. Unlike conventional ensembles where models operate independently, each model $m_i$ directly accesses and fuses the hidden representations of its predecessor $m_{i-1}$, thereby enabling hierarchical interaction across the $n$ LLMs. Concretely, at the $t$-th decoding step, let $h_{l-1,t}^{(i-1)}$

denote the hidden state from the $(l-1)$-th Transformer layer of the $(i-1)$-th model. The cross-model attention mechanism at the $l$-th layer of the $i$-th model is defined as:

$$\tilde{h}_{l,t}^{(i)} = \begin{cases} \text{LayerNorm}\big(h_{l-1,t}^{(i)} + h_{l-1,t}^{(i-1)}\big), & \text{if } l \bmod \eta = 0, \\ h_{l-1,t}^{(i)}, & \text{otherwise.} \end{cases} \tag{1}$$

where $\eta$ is an integer constant that controls the sparsity of cross-model connections among the $n$ LLMs. With the enhanced hidden states, the cross-attention output is calculated by:

$$\hat{h}_{l,t}^{(i)} = \text{softmax}\Big(\frac{Q_{l,t}^{(i)}(K_{l,t}^{(i)})^\top}{\sqrt{d_k}}\Big)V_{l,t}^{(i)}, \ \text{where } Q_{l,t}^{(i)} = \tilde{h}_{l,t}^{(i)}W_Q, \quad K_{l,t}^{(i)} = \tilde{h}_{l,t}^{(i)}W_K, \quad V_{l,t}^{(i)} = \tilde{h}_{l,t}^{(i)}W_V, \tag{2}$$

followed by another residual enhancement:

$$h_{l,t}^{(i)} = \text{LayerNorm}\big(\hat{h}_{l,t}^{(i)} + \tilde{h}_{l,t}^{(i)}\big). \tag{3}$$

This design equips later models with structured access to their predecessors' reasoning process, yielding: *error correction*, where inaccurate states are adjusted, and *knowledge reinforcement*, where correct signals are amplified.

For the input and output streaming, the input sequence $X_\tau$ is shared across all models in LLM-BOOST, while the output fusion streaming incorporates two key innovations.

(1) **Error-Token Forward Pass**. To enable successor models to recognize and correct token-level errors made by their predecessors, we introduce an error-token forwarding mechanism. This design improves both performance and robustness of LLMBOOST. During training, the error logits produced by model $m_i$ are sequentially passed to its successor $m_{i+1}$ under a specific learning objective (detailed in the next section). At the $t$-th decoding step, let $\hat{y}_t^{(i)} = \arg\max p_t^{(i)}$ denote the predicted token index, where the error tokens are defined as follows:

$$\text{ErrorToken}_t^{(i)} = \begin{cases} \hat{y}_t^{(i)}, & \text{if } \hat{y}_t^{(i)} \neq y_t^*, \\ \emptyset, & \text{otherwise,} \end{cases} \tag{4}$$

where $y_t^*$ denotes the ground-truth token. Thus, only incorrect token indices are passed forward, encouraging successors to explicitly learn from predecessor mistakes.

(2) **Key-Logits Backward Pass**. To further stabilize the inference process, we propose a top-$k$ logits fusion strategy, where later successor models pass key logits backward to the base model $m_0$. Instead of aggregating over the entire vocabulary, each sub-model contributes only its most informative token logits, thereby reducing the influence of noisy, low-confidence outputs. Formally, given the logit distribution $z_t^{(i)}$ from the $i$-th model $m_i$, we define the restricted candidate set and the ensemble update rule as:

$$z_t = z_t^{(0)} + \sum_{i=1}^{n} \lambda_i \cdot \text{Mask}(z_t^{(i)}, \mathcal{V}_{\text{top-}k}^{(i)}), \ \text{where } \mathcal{V}_{\text{top-}k}^{(i)} = \arg\text{top-k}(z_t^{(i)}), \tag{5}$$

where $z_t$ denotes the final raw logits, $\lambda_i$ is a scaling coefficient, and $\text{Mask}(\cdot)$ restricts contributions to the top-$k$ candidates by setting all other indices to zero in each logit distribution.

## 3.2 CHAIN TRAINING PARADIGM

**Error-Suppression Objective.** To enable each model to refine its predecessor by correcting erroneous predictions while preserving correct ones, we introduce a token-level suppression loss. The principle is that each $m_i$ should preserve plausible predictions from $m_{i-1}$, while intervening only when necessary. Formally, we define the error-suppression loss:

$$\mathcal{L}_{s,t} = -\log\Big(\sigma\big(\beta\left[\log p_t^{(i)}[y_t^*] - \log p_t^{(i)}[\text{ErrorToken}_t^{(i-1)}]\right]\big)\Big) \cdot \mathbf{1}\{\text{ErrorToken}_t^{(i-1)} \neq \emptyset\}, \tag{6}$$

where $\sigma$ is the sigmoid, $\beta$ a scaling factor, and the indicator $\mathbf{1}$ activates the loss only at positions where the predecessor $i-1$ predicts incorrectly. The overall objective for $m_i$ becomes:

$$\mathcal{L}_i = \sum_t \mathcal{L}_{s,t} - \alpha \sum_t \log p_t^{(i)}[y_t^*], \tag{7}$$

where the right item is the cross-entropy loss, and $\alpha$ is a balance coefficient. This dual-objective strategy endows the ensemble with a *correction-while-retaining* ability, preserving useful signals while refining erroneous ones.

---

**Algorithm 1:** LLMBOOST: Training Process

**Input:** Base model list $\mathcal{M} = \{m_0, m_1, \ldots, m_n\}$, training dataset $\mathcal{D}$,
**Output:** Fine-tuned model list $\tilde{\mathcal{M}} = \{\tilde{m}_0, \tilde{m}_1, \ldots, \tilde{m}_n\}$

1  Initialize log pool $g$ ;
2  **Stage 1: Train the base model $m_0$;**
3  **for** *each batch* $(x, y^*) \in \mathcal{D}$ **do**
4      Forward pass: $z^{(0)} \leftarrow m_0(x)$;
5      Compute Cross-Entropy loss:$\mathcal{L}_0 \leftarrow \text{CrossEntropyLoss}(z_0, y^*)$;
6      Update $m_0$ parameters via LoRA using $\mathcal{L}_0$;
7  **end**
8  Obtain fine-tuned model: $\tilde{m}_0 \leftarrow m_0$;
9  **Stage 2: Train subsequent models $\{m_1(\cdot; \theta_1), \ldots, m_n(\cdot; \theta_n)\}$;**
10 **for** $i = 1$ **to** $n$ **do**
11     **for** *each batch* $(x, y^*) \in \mathcal{D}$ **do**
12         Get previous model's inference trajectory: $\{h^{(i-1)}, \text{ErrorToken}^{(i-1)}\} \leftarrow \tilde{m}_{i-1}(x)$;
13         Forward pass: $z^{(i)} \leftarrow m_i(x, h^{(i-1)})$;
14         Compute Error-Suppression loss $\mathcal{L}_i$ via 7 using $\{z^{(i)}, y^*, \text{ErrorToken}^{(i-1)}\}$;
15         Update $m_i$ parameters via LoRA using $\mathcal{L}_i$;
16     **end**
17     Obtain fine-tuned model: $\tilde{m}_i \leftarrow m_i$;
18 **end**
19 **return** $\{\tilde{m}_0(\cdot; \theta_0), \tilde{m}_1(\cdot; \theta_1), \ldots, \tilde{m}_n(\cdot; \theta_n)\}$

---

**Training Pipeline**. Our model adopts a two-stage chained fine-tuning strategy (see Algorithm 1), which ensures that each sub-model learns not only from ground-truth supervision but also from the reasoning trajectory of its predecessor. In the first stage, the base model $m_0$ is adapted to the target task following standard LoRA fine-tuning. The parameter updates of $m_0$ are solely driven by the input data $x$ and the ground-truth labels $y^*$. In the second stage, subsequent models $m_i$ are fine-tuned in a chained manner. Specifically, for each model, we first extract the internal hidden states $h^{(i-1)}$ and error tokens $\text{ErrorToken}^{(i-1)}$ from its predecessor $\tilde{m}_{i-1}$. The input $x$, together with $h^{(i-1)}$, is then fed into $m_i$ for training. The Error-Suppression loss is computed using $z^{(i)}$, $y^*$, and $\text{ErrorToken}^{(i-1)}$. This mechanism enables later models to progressively correct the reasoning errors of earlier models, thereby improving the overall ensemble performance.

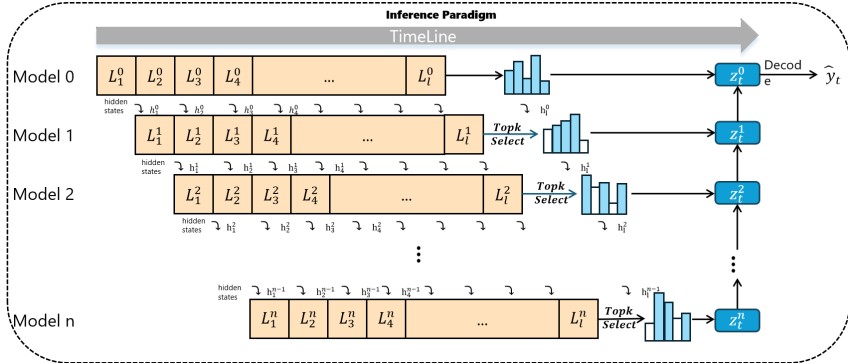

Figure 2: **Near-parallel inference paradigm.** The figure uses the timeline as the axis to show the computation state of each model at each moment. Hidden states are propagated across layers and models, and logits are aggregated and decoded into the final prediction, enabling overlapped computation and reduced inference latency.

## 3.3 NEAR-PARALLEL INFERENCE PARADIGM

As each successor model $m_i$ depends on the intermediate hidden states of its predecessor $m_{i-1}$, a naive decoding scheme requires $m_i$ to begin its decoding only after $m_{i-1}$ has completed the entire decoding process. This results in an inference time complexity of $O(n)$ if one model takes $O(1)$ time per sequence, which becomes impractical when $n$ is large.

To address this, we propose a near-parallel inference paradigm (Algorithm 2 in Appendix E), which enables efficient inference across multiple models. Instead of waiting for the predecessor to finish full sequence generation, LLMBOOST streams hidden states at the *layer level*: the hidden state from the $(l-1)$-th layer of $m_{i-1}$ is immediately passed to the $l$-th layer of $m_i$ while the forward pass of $m_{i-1}$ is still ongoing (see Figure 2). For implementation, we maintain a thread-safe shared hidden state pool $g$ for inter-model communication. At each decoding step $t$ and each layer $l$, model $m_i$ waits for the corresponding hidden state $h_{l-1,t}^{(i-1)}$ from its upstream model $m_{i-1}$. Once $h_{l-1,t}^{(i-1)}$ becomes available, it is pushed into $g$ and immediately consumed by $m_i$ to compute $h_{l,t}^{(i)}$.

Finally, after all $(n+1)$ models complete their forward passes at step $t$, their logits are fused following the Key-Logits Backward Pass rule. The final logits $z_t$ are computed according to Eq. 5, from which the next token is generated.

# 4 THEORETICAL GUARANTEES: WHY SEQUENTIAL CORRECTION WORKS

In the previous section, we introduced LLMBOOST, a framework built on the principle of sequential error correction. A natural and critical question arises: does this boosting-style integration of models guarantee a performance improvement? This section provides a formal theoretical justification, demonstrating that under a practical condition, adding a new model to the ensemble provably reduces the prediction error.

Our analysis focuses on the Mean Squared Error (MSE) between the ensemble's predicted probability distribution and the one-hot ground truth at each auto-regressive decoding step. The core of our theory hinges on formalizing the intuitive idea that each successor model $m_i$ is trained to correct the residual errors of the preceding ensemble $\mathcal{M}_{i-1} = \{m_0, \ldots, m_{i-1}\}$. This corrective capability is primarily learned through the error-suppression objective (Eq. 7) and enabled by the cross-model attention mechanism. We encapsulate this requirement in a single, core assumption. For brevity, we present the main results here; the full formal setup and detailed proofs are provided in Appendix I.

**Notations for Analysis**. At decoding step $t$, let $A_i$ denote the sampling space of $m_i$. We consider an ensemble $\mathcal{M}_i$ with accumulated logits $\hat{z}_t^{(i)}$, yielding predicted distribution $\hat{P}_t^{(i)} = \text{softmax}(\hat{z}_t^{(i)})$. Let $\hat{y}_{<t}^{(i-1)}$ be the sequence generated by $\mathcal{M}_{i-1}$ up to step $t - 1$, and let $p_t^*(\cdot \mid X_\tau, \hat{y}_{<t}^{(i-1)})$ denote the ground-truth distribution, represented by the one-hot vector $p_t^*$. The residual error of $\mathcal{M}_{i-1}$ is defined as $e_t^{(i-1)} = p_t^* - \hat{P}_t^{(i-1)}$. To quantify how a new model $m_i$ influences the ensemble's prediction, we define its effective linear contribution to the probability space as $g_t^{(i)} \triangleq J_{\text{softmax}}(\hat{z}_t^{(i-1)})z_t^{(i)}$, where $J_{\text{softmax}}(\hat{z}_t^{(i-1)})$ is the Jacobian matrix of the softmax function evaluated at $\hat{z}_t^{(i-1)}$.

**Assumption 1.** *For any model $i \geq 1$, any decoding step $t$, and any vocabulary dimension $v \in \{1, \ldots, |\mathcal{V}|\}$, we assume that the successor model $m_i$ produces an effective linear contribution $g_t^{(i)}$ that is a valid corrective term of the residual error vector $e_t^{(i-1)}$. Specifically, we assume its systematic bias in fitting this residual is less than the total uncertainty of the residual. Formally, given a generated context $(X_\tau, \hat{y}_{<t}^{(i-1)})$:*

$$\epsilon_{g_{t,v}^{(i)}} < \sqrt{\epsilon_{i-1,t,v}^2 + \sigma_{i-1,t,v}^2}, \tag{8}$$

*where we let $\Theta_{i-1} := \{\theta_0, \ldots, \theta_{i-1}\}$ be the parameters of the ensemble $\mathcal{M}_{i-1}$, drawn from a joint distribution $\mathcal{A}_{i-1}$. The bias ($\epsilon^2$) and variance ($\sigma^2$) terms are defined as:*

$$\epsilon_{i-1,t,v}^2 := \mathbb{E}_{(X_\tau,Y_\tau)\sim\mathcal{D}} \left[ \left( p_{t,v}^* - \mathbb{E}_{\Theta_{i-1}\sim\mathcal{A}_{i-1}}[\hat{P}_{t,v}^{(i-1)} \mid X_\tau, \hat{y}_{<t}^{(i-1)}] \right)^2 \right] < \infty, \qquad (9)$$

$$\sigma_{i-1,t,v}^2 := \mathbb{E}_{(X_\tau,Y_\tau)\sim\mathcal{D}} \left[ \text{Var}_{\Theta_{i-1}\sim\mathcal{A}_{i-1}}[\hat{P}_{t,v}^{(i-1)} \mid X_\tau, \hat{y}_{<t}^{(i-1)}] \right] < \infty, \qquad (10)$$

$$\epsilon_{g_{t,v}^{(i)}}^2 := \mathbb{E}_{\substack{\Theta_{i-1}\sim\mathcal{A}_{i-1} \\ (X_\tau,Y_\tau)\sim\mathcal{D}}} \left[ \left( e_{t,v}^{(i-1)} - \mathbb{E}_{\theta_i\sim A_i}[g_{t,v}^{(i)} \mid \Theta_{i-1}, X_\tau, \hat{y}_{<t}^{(i-1)}] \right)^2 \right] < \infty. \qquad (11)$$

*Furthermore, we assume that the logits generated by the successor model have a finite fourth moment, i.e.,* $\mathbb{E}[\|z_t^{(i)}(\cdot \mid X_\tau, \hat{y}_{<t}^{(i-1)})\|_2^4] < \infty.$

This assumption posits that each boosting step is productive—the new model is more accurate at predicting the error than the existing ensemble is at predicting the target. Our chain training paradigm is explicitly designed to encourage this condition. Based on this assumption, we establish the following theorem and corollary.

**Theorem 1.** *Under Assumption 1, For each model $i \geq 1$, decoding step $t$, and vocabulary dimension $v$, there exists an upper bound $\lambda_{i,t,v}^* > 0$ for the scaling coefficient. For any weight $\lambda_i \in (0, \lambda_{i,t,v}^*)$, the expected squared error of the new ensemble $\mathcal{M}_i$ at inference is strictly lower than that of the predecessor ensemble $\mathcal{M}_{i-1}$ for that dimension. Specifically:*

$$\mathbb{E}_{\substack{\Theta_{i-1}\sim\mathcal{A}_{i-1} \\ \theta_i\sim A_i \\ (X_\tau,Y_\tau)\sim\mathcal{D}}} \left[ \left( p_{t,v}^* - \hat{P}_{t,v}^{(i)} \right)^2 \mid X_\tau, \hat{y}_{<t}^{(i-1)} \right] < \mathbb{E}_{\substack{\Theta_{i-1}\sim\mathcal{A}_{i-1} \\ (X_\tau,Y_\tau)\sim\mathcal{D}}} \left[ \left( p_{t,v}^* - \hat{P}_{t,v}^{(i-1)} \right)^2 \mid X_\tau, \hat{y}_{<t}^{(i-1)} \right].$$

$$(12)$$

**Corollary 1.** *For each model $i \geq 1$ and decoding step $t$, there exists a single, per-token scalar $\lambda_{i,t}^* > 0$, independent of vocabulary dimension, such that for any scaling coefficient $\lambda_i \in (0, \lambda_{i,t}^*)$, the total Mean Squared Error of the new ensemble $\mathcal{M}_i$ for that token prediction at inference is strictly less than that of the predecessor ensemble $\mathcal{M}_{i-1}$:*

$$\mathbb{E}_{\substack{\Theta_{i-1}\sim\mathcal{A}_{i-1} \\ \theta_i\sim A_i \\ (X_\tau,Y_\tau)\sim\mathcal{D}}} \left[ \|p_t^* - \hat{P}_t^{(i)}\|_2^2 \mid X_\tau, \hat{y}_{<t}^{(i-1)} \right] < \mathbb{E}_{\substack{\Theta_{i-1}\sim\mathcal{A}_{i-1} \\ (X_\tau,Y_\tau)\sim\mathcal{D}}} \left[ \|p_t^* - \hat{P}_t^{(i-1)}\|_2^2 \mid X_\tau, \hat{y}_{<t}^{(i-1)} \right]. \quad (13)$$

**Remark and Interpretation**. Together, Theorem 1 and Corollary 1 provide a principled guarantee for the progressive improvement of LLMBOOST. They establish that as long as each new model serves as a valid corrector (Assumption 1), the overall system's inference accuracy is guaranteed to improve. The existence of an upper bound $\lambda_i^*$ highlights a classic bias-variance trade-off: a larger $\lambda_i$ incorporates more of the new model's corrective signal (reducing bias), but also more of its inherent instability (adding variance). Our framework thus provides a theoretically sound path for enhancing model accuracy. We provide an optimization-based analysis in Appendix J to argue why our training objective naturally leads to models that satisfy Assumption 1. These theoretical findings set the stage for our empirical evaluation, where we will demonstrate the practical performance gains in Section 5.

## 5 EXPERIMENTS

We conduct extensive experiments to evaluate the effectiveness and efficiency of our proposed method. First, we evaluate LLMBOOST's performance on two representative tasks, Commonsense Reasoning and Arithmetic Reasoning. Second, to verify the generality of our approach, we further evaluate LLMBOOST in an industrial scenario provided by the *a specific laboratory*. Third, we evaluate the time efficiency of our near-parallel inference paradigm. Additional experimental results and ablation studies are included in Appendix H.

**Commonsense & Arithmetic Datasets.** To comprehensively evaluate LLMBOOST, we utilize a broad suite of reasoning tasks: (i) Commonsense reasoning: PIQA (Bisk et al., 2020), HellaSwag (Zellers et al., 2019), WinoGrande (Sakaguchi et al., 2021),BoolQ (Clark et al., 2019), SIQA (Sap et al., 2019), and OpenbookQA (OBQA) (Mihaylov et al., 2018). (ii) Arithmetic reasoning: AQuA (Ling et al., 2017),GSM8K (Cobbe et al., 2021), MAWPS (Koncel-Kedziorski et al., 2016), and

Table 1: Comparison of commonsense and arithmetic reasoning performance under matched parameter scales. Results show that LLMBOOST consistently outperforms competitive baselines across different model families and sizes, highlighting its robustness and effectiveness in improving reasoning accuracy.

| Model | Params | Commonsense Reasoning (Acc. ↑) | | | | | | | Arithmetic Reasoning (Acc. ↑) | | | | |
|---|---|---|---|---|---|---|---|---|---|---|---|---|---|
| | | PIQA | WinoG. | HellaS. | BoolQ | SIOA | OBQA | Avg. | AQuA | GSM8K | MAWPS | SVAMP | Avg. |
| **Llama-3.1-8B** | $1 \times 8B$ | 83.3 | 81.8 | 90.6 | 69.5 | 75.8 | 79.0 | 80.0 | 42.3 | 63.7 | 89.5 | 77.4 | 68.2 |
| T-copilot | $2 \times 8B$ | 84.2 | 83.2 | 91.4 | 69.8 | 76.9 | 78.4 | 81.0 | 42.8 | 66.5 | 89.7 | 75.7 | 68.7 |
| vote | $2 \times 8B$ | 81.7 | 81.3 | 86.8 | 67.3 | 76.1 | 84.8 | 79.7 | 44.4 | 67.8 | 89.5 | 77.5 | 69.8 |
| UNITE | $2 \times 8B$ | 86.9 | 84.9 | 93.5 | 70.5 | 79.2 | 84.8 | 83.3 | 42.8 | 65.5 | 89.3 | 76.5 | 68.5 |
| LLMBOOST | $2 \times 8B$ | 87.9 | 86.1 | 94.5 | 71.7 | 81.6 | 85.6 | 84.4 | 46.8 | 68.8 | 90.1 | 80.1 | 71.5 |
| vote | $3 \times 8B$ | 87.7 | 83.2 | 90.3 | 71.8 | 80.4 | 85.4 | 83.1 | 43.9 | 66.0 | 90.7 | 78.1 | 69.7 |
| UNITE | $3 \times 8B$ | 85.3 | 85.0 | 92.2 | 70.9 | 79.9 | 85.0 | 83.1 | 43.3 | 65.6 | 88.3 | 77.2 | 68.6 |
| LLMBOOST | $3 \times 8B$ | 87.4 | 86.3 | 95.2 | 72.5 | 80.7 | 84.8 | **84.5** | 47.6 | 68.5 | 92.0 | 81.3 | **72.4** |
| **LLama-3.2-3B** | $1 \times 3B$ | 78.5 | 77.7 | 83.9 | 62.4 | 74.3 | 76.8 | 75.6 | 31.9 | 49.7 | 84.9 | 65.8 | 58.1 |
| T-copilot | $2 \times 3B$ | 75.7 | 75.9 | 83.4 | 64.6 | 73.8 | 77.2 | 75.1 | 33.3 | 49.5 | 84.6 | 66.7 | 58.5 |
| vote | $2 \times 3B$ | 79.4 | 76.9 | 84.7 | 66.2 | 75.4 | 75.8 | 76.4 | 34.6 | 50.9 | 85.4 | 68.2 | 59.8 |
| UNITE | $2 \times 3B$ | 82.8 | 80.0 | 88.7 | 66.5 | 76.7 | 77.4 | 78.7 | 31.1 | 50.9 | 85.2 | 65.7 | 58.2 |
| LLMBOOST | $2 \times 3B$ | 84.7 | 81.4 | 90.4 | 67.1 | 78.0 | 78.8 | 80.1 | 35.8 | 51.5 | 87.4 | 70.1 | 61.2 |
| vote | $3 \times 3B$ | 80.5 | 76.4 | 85.1 | 66.7 | 77.5 | 77.2 | 77.2 | 32.9 | 53.5 | 87.4 | 66.4 | 60.1 |
| UNITE | $3 \times 3B$ | 82.8 | 81.3 | 89.9 | 67.1 | 78.4 | 78.2 | 79.6 | 32.5 | 52.9 | 86.0 | 66.0 | 59.4 |
| LLMBOOST | $3 \times 3B$ | 84.2 | 82.0 | 91.5 | 70.2 | 79.0 | 79.6 | **81.1** | 35.1 | 54.3 | 86.2 | 69.9 | **61.4** |
| **Qwen-2.5-7B** | $1 \times 7B$ | 86.7 | 83.9 | 89.4 | 68.7 | 77.6 | 87.4 | 82.3 | 56.8 | 75.1 | 92.0 | 86.1 | 77.5 |
| T-copilot | $2 \times 7B$ | 87.1 | 84.3 | 90.2 | 70.8 | 79.3 | 89.2 | 83.5 | 56.9 | 75.6 | 92.7 | 85.3 | 77.6 |
| vote | $2 \times 7B$ | 86.3 | 84.5 | 91.3 | 70.7 | 80.0 | 88.8 | 83.6 | 57.8 | 76.3 | 91.6 | 86.4 | 78.0 |
| UNITE | $2 \times 7B$ | 86.1 | 84.1 | 91.6 | 68.6 | 79.1 | 88.2 | 83.0 | 59.5 | 76.3 | 92.2 | 86.4 | 78.6 |
| LLMBOOST | $2 \times 7B$ | 89.5 | 86.7 | 95.1 | 71.6 | 79.1 | 91.6 | 85.6 | 59.8 | 78.1 | 94.1 | 88.8 | 80.2 |
| vote | $3 \times 7B$ | 87.5 | 82.2 | 91.3 | 70.2 | 79.6 | 85.8 | 82.7 | 60.9 | 77.4 | 92.4 | 86.7 | 79.4 |
| UNITE | $3 \times 7B$ | 86.1 | 83.3 | 91.5 | 70.3 | 78.4 | 86.8 | 82.7 | 60.9 | 77.8 | 92.8 | 86.3 | 79.5 |
| LLMBOOST | $3 \times 7B$ | 89.9 | 88.8 | 95.4 | 74.4 | 81.2 | 91.4 | **86.9** | 61.8 | 78.9 | 93.3 | 88.4 | **80.6** |
| **Qwen-2.5-3B** | $1 \times 3B$ | 84.5 | 80.1 | 80.6 | 68.1 | 77.7 | 84.6 | 79.3 | 54.1 | 69.5 | 89.9 | 83.4 | 74.2 |
| T-copilot | $2 \times 3B$ | 83.9 | 80.2 | 82.6 | 68.9 | 78.4 | 86.4 | 80.1 | 53.9 | 72.1 | 90.1 | 82.7 | 74.7 |
| vote | $2 \times 3B$ | 83.4 | 80.0 | 86.3 | 68.7 | 77.9 | 84.0 | 80.1 | 54.2 | 71.2 | 89.5 | 82.6 | 74.4 |
| UNITE | $2 \times 3B$ | 84.1 | 80.6 | 83.5 | 67.3 | 78.0 | 83.0 | 79.4 | 58.5 | 71.6 | 91.3 | 84.7 | 76.5 |
| LLMBOOST | $2 \times 3B$ | 85.9 | 81.8 | 91.4 | 69.7 | 79.0 | 86.2 | 82.3 | 60.2 | 74.0 | 92.4 | 85.3 | 78.0 |
| vote | $3 \times 3B$ | 84.1 | 79.7 | 87.6 | 67.8 | 77.4 | 83.8 | 80.4 | 59.7 | 71.9 | 92.1 | 85.2 | 77.2 |
| UNITE | $3 \times 3B$ | 84.8 | 80.8 | 85.1 | 66.2 | 77.3 | 86.0 | 80.0 | 59.3 | 71.6 | 92.3 | 86.0 | 77.3 |
| LLMBOOST | $3 \times 3B$ | 86.1 | 82.0 | 91.7 | 70.4 | 80.0 | 87.6 | **83.0** | 61.2 | 73.4 | 92.6 | 85.6 | **78.2** |

Table 2: Performance comparison (%) across both matched-scale ensembles and weakly-heterogeneous ensembles.

| Model | Params | Latency (s/token ↓) | Commonsense Reasoning (Acc. ↑) | | | | | | | Arithmetic Reasoning (Acc. ↑) | | | | |
|---|---|---|---|---|---|---|---|---|---|---|---|---|---|---|
| | | | PIQA | WinoG. | HellaS. | BoolQ | SIOA | OBQA | Avg. | AQuA | GSM8K | MAWPS | SVAMP | Avg. |
| Qwen2.5-7B | 7B | 0.056 | 86.7 | 83.9 | 89.4 | 68.7 | 77.6 | 87.4 | 82.3 | 56.8 | 75.1 | 92.0 | 86.1 | 77.5 |
| LLMBOOST (3B × 2) | 6B (-1B) | **0.043** | 85.9 | 81.8 | 91.7 | 70.4 | 80.0 | 86.2 | 82.3 | 60.2 | 74.0 | 92.4 | 85.3 | **78.0** |
| Qwen2.5-14B | 14B | 0.098 | 90.2 | 85.8 | 92.9 | 71.2 | 78.9 | 91.4 | 85.1 | 62.2 | 78.5 | 92.4 | 87.4 | 80.1 |
| LLMBOOST (7B × 2) | 14B | **0.075** | 89.5 | 86.7 | 95.1 | 71.6 | 79.1 | 91.6 | **85.6** | 59.8 | 78.1 | 94.1 | 88.8 | **80.2** |
| ***Weakly-Heterogeneous Ensemble*** | | | | | | | | | | | | | | |
| LLama-3.2-3B | 3B | 0.027 | 78.5 | 77.7 | 83.9 | 62.4 | 74.3 | 76.8 | 75.6 | 31.9 | 49.7 | 84.9 | 65.8 | 58.1 |
| Llama-3.1-8B | 8B | 0.038 | 83.3 | 81.8 | 90.6 | 69.5 | 75.8 | 79.0 | 80.0 | 42.3 | 63.7 | 89.5 | 77.4 | 68.2 |
| LLMBOOST (8B + 3B) | 11B | 0.049 | 85.9 | 85.0 | 93.8 | 71.3 | 80.4 | 84.0 | **83.7** | 43.7 | 64.4 | 89.7 | 78.6 | **69.1** |

SVAMP (Patel et al., 2021). More details can be found in Appendix D.1.

**Toolchain Dataset(TCD).** We construct a dataset from real-world public cloud scenarios in a specific laboratory, targeting the evaluation of AI agents in toolchain scheduling. It assesses whether LLMs can interpret user intentions, plan multi-step toolchains, and execute tool calls under realistic industrial constraints. More details can be found in Appendix D.1.

**Baselines.** To clearly validate the effectiveness of the proposed method, experiments compare it with three mainstream large model optimization and ensembling approaches, namely VOTE (Li et al., 2024a), UNITE (Yao et al., 2024), and T-copilot (Zou et al., 2025). Detailed description of these three baseline methods can be found in Appendix D.2.

**Implementation Details.** For LLMBOOST, we employ models from both the LLaMA-3 family (LLaMA-3.2-3B, LLaMA-3.1-8B) and the Qwen-2.5 family (Qwen-2.5-3B, Qwen-2.5-7B) as sub-models. We further extend the generation function from the Huggingface Transformers library to support distributed, token-level inference acceleration across multiple sub-models. All experiments are conducted on four NVIDIA H800 GPUs. More details can be found in Appendix H.1.

Table 3: Accuracy (%) across different ensemble sizes on the arithmetic reasoning dataset.

| Model | Params | AQuA | GSM8K | MAWPS | SVAMP | Avg. |
|---|---|---|---|---|---|---|
| **Llama-3.1-8B** | | | | | | |
| LLMBOOST | $1 \times 8B$ | 42.3 | 63.7 | 89.5 | 77.4 | 68.2 |
| LLMBOOST | $2 \times 8B$ | 46.8 | 68.8 | 90.1 | 80.1 | 71.5 |
| LLMBOOST | $3 \times 8B$ | 47.6 | 68.5 | 92.0 | 81.3 | 72.4 |
| LLMBOOST | $4 \times 8B$ | 47.1 | 69.3 | 92.1 | 80.9 | 72.4 |
| **Qwen2.5-7B** | | | | | | |
| LLMBOOST | $1 \times 7B$ | 56.8 | 75.1 | 92.0 | 86.1 | 77.5 |
| LLMBOOST | $2 \times 7B$ | 59.8 | 78.1 | 94.1 | 88.8 | 80.2 |
| LLMBOOST | $3 \times 7B$ | 61.8 | 78.9 | 93.2 | 88.4 | 80.6 |
| LLMBOOST | $4 \times 7B$ | 62.2 | 78.7 | 93.5 | 88.5 | 80.7 |

Table 4: Accuracy (%) on Toolchain Dataset.

| Model | Params | TCD |
|---|---|---|
| **Llama-3.1-8B** | | |
| Llama | $1 \times 8B$ | 52.0 |
| VOTE | $2 \times 8B$ | 53.0 |
| UNITE | $2 \times 8B$ | 52.5 |
| LLMBOOST | $2 \times 8B$ | **55.0** |
| **Qwen-2.5-7B** | | |
| Qwen | $1 \times 7B$ | 41.0 |
| VOTE | $2 \times 7B$ | 43.0 |
| UNITE | $2 \times 7B$ | 41.5 |
| LLMBOOST | $2 \times 7B$ | **45.5** |

## 5.1 MAIN EXPERIMENTS RESULTS

Table 1 reports LLMBOOST 's performance against ensemble baselines on commonsense and arithmetic reasoning tasks, as well as the WinoG dataset. Compared with voting ensembles, LLMBOOST yields notable improvements: up to 5.9% for commonsense reasoning and 3.9% for arithmetic reasoning. Against stronger baselines, it still achieves consistent gains of up to 1.7% and 3.9% on the two tasks, respectively. Scaling from a single model to two and three models further amplifies performance, with commonsense accuracy rising by 5.5% and arithmetic by 6.2%. On WinoG, LLMBOOST delivers its largest improvement, surpassing the best baseline by 6.6%. Overall, the results confirm that sequential ensembling with LLMBOOST reliably strengthens reasoning ability across tasks and model scales.

**Comparison under equal parameter budgets.** To control for model size, we compare LLM-BOOST with single-model baselines under the same total parameters. As shown in Table 2, LLM-BOOST built from two Qwen2.5-3B models (6B total) reaches the same level of commonsense performance as the 7B model and achieves a higher arithmetic score, improving the average from 77.5 to 78.0, while using 1B fewer parameters. Under a 14B budget, LLMBOOST constructed from two Qwen2.5-7B models also shows consistent advantages. On HellaSwag, performance increases from 92.9 to 95.1; on SVAMP, from 87.4 to 88.8; and the overall commonsense average rises from 85.1 to 85.6. Arithmetic reasoning similarly improves from 80.1 to 80.2. In terms of efficiency, per-token latency decreases from 98 ms to 75 ms, corresponding to a 23% speedup under the same parameter budget. Overall, LLMBOOST provides both higher accuracy and faster inference than single LLMs with equivalent total size.

**Weakly Heterogeneous Model Ensemble.** We further evaluate LLMBOOST under a weakly heterogeneous configuration by integrating two models from the same family but belonging to different series. In practice, integrating models from different families is difficult because their tokenizers are incompatible; even within a single family, different parameter sizes may adopt different vocabularies (e.g., Qwen2.5-7B and Qwen2.5-3B). Such tokenizer mismatches lead to inconsistent token boundaries during decoding, preventing LLMBOOST from performing accurate layer-wise residual correction. Based on this, we focus on the compatible model pair of Llama3.1-8B and Llama3.2-3B, which maintain consistency in both tokenizer and architectural specifications. As shown in Table 2, the weakly heterogeneous ensemble (8B + 3B) improves the average commonsense score from 80.0 to 83.7 and raises the arithmetic average from 68.2 to 69.1 compared with the stronger 8B model alone. This accuracy gain comes with only a minimal increase in computational cost: the per-token latency rises slightly from 0.038 s to 0.049 s, indicating that the ensemble adds only marginal overhead. These results show that even when combining models of different series and scales, LLM-BOOST continues to provide consistent benefits in both accuracy and efficiency. This demonstrates that our boosting framework extends beyond strictly homogeneous ensembles and offers a viable approach for integrating diverse model variants within the same family.

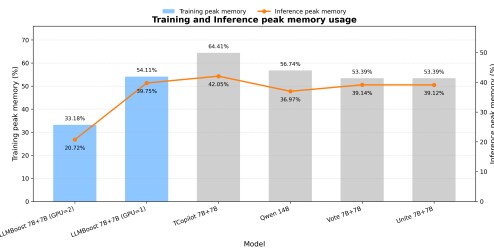

(a) Training and inference peak memory usage across different models.

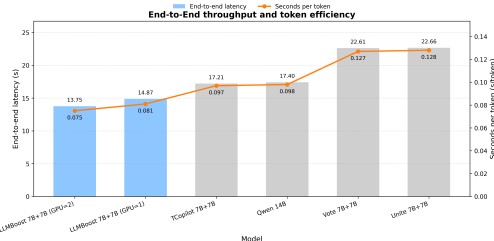

(b) End-to-end latency and per-token generation cost.

Figure 3: Overall system efficiency comparison of LLMBOOST and baseline ensemble methods.

**Performance on TCD.** Table 4 reports the performance on Toolchain Dataset.LLMBOOST consistently surpasses both individual models and other ensemble baselines (VOTE, UNITE) across the Llama-3.1-8B and Qwen-2.5-7B models.Specifically, it improves over the strongest baseline by 3% with Llama-3.1-8B and 2.5% with Qwen-2.5-7B. Moreover, in internal end-to-end toolchain scheduling tests within a specific laboratory, our method further achieves a gain of 5%, demonstrating strong practical utility in complex production workflows.

**Study on Number of LLMs.** We further explore the impact of varying the number of sub-models $n$ in LLMBOOST. As shown in Table 3, performance improves consistently when increasing $n$ from 1 to 4. This monotonic trend confirms that subsequent models are able to refine and strengthen the outputs of their predecessors. However, the improvement exhibits diminishing returns beyond a certain number of sub-models ($n \geq 3$), implying that most of the corrective and reinforcing capacity is already realized in the earlier stages of the ensemble. In addition, increasing $n$ inevitably leads to higher computational and latency costs. By experimental results, we observe that $n = 3$ achieves an effective balance between accuracy and efficiency, yielding strong gains without excessive overhead. Overall, these findings demonstrate that LLMBOOST can achieve substantial benefits even with a relatively small ensemble size.

**Efficiency and Cost Analysis.** We evaluate end-to-end wall-clock time, per-token latency, and peak GPU memory during both training and inference, with results shown in Figure 3. LLMBOOST (7B+7B, near-parallel, 2 GPUs) achieves an end-to-end latency of 13.75 seconds, significantly outperforming ensemble baselines such as UNITE (22.66 s), VOTE (22.61 s), and T-Copilot (17.21 s), and it remains competitive even when restricted to a single GPU. The per-token latency of LLMBOOST (0.075 s/token) is also lower than that of VOTE (0.127 s/token), UNITE (0.128 s/token), and T-Copilot (0.097 s/token), reflecting improved throughput. In terms of memory usage, LLMBOOST reaches 33.18% peak training memory and 20.72% peak inference memory per GPU in the 2-GPU configuration, which is comparable to or below the levels of other ensemble systems. The 1-GPU results follow a similar trend. The cost introduced by cross-model state passing is minimal 0.0009 s on 2 GPUs and 0.0002 s on 1 GPU and has negligible impact on the overall latency. Taken together, these findings show that LLMBOOST achieves a more favorable combination of speed, memory efficiency, and coordination cost than existing ensemble approaches, while still providing strong accuracy gains. All efficiency measurements were conducted on an NVIDIA H800 GPU cluster using the AQuA dataset, with identical decoding settings across all methods. A theoretical analysis of inference-time behavior under different execution strategies is provided in Appendix G.

# 6 CONCLUSION

We propose LLMBOOST, a novel boosting-style ensemble learning framework for LLMs. It leverages cross-model attention, enabling subsequent models to utilize the intermediate hidden states of their predecessors. We further introduce a near-parallel inference paradigm that substantially improves efficiency. Extensive experiments demonstrate that LLMBOOST outperforms baselines such as VOTE, effectively enhancing reasoning ability and significantly accelerating inference compared to traditional ensembles. Additional discussions are provided in Appendix C.

**Ethics Statement**

All experiments use publicly available or anonymized synthetic datasets, with generated content verified for safety. All authors declare no conflicts of interest, and relevant code/data are open-source for transparency.

**Reprodicibility Statement**

We provide open-source code with detailed docs.Following these materials, all reported experimental results can be fully replicated.

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

# Appendix

CONTENTS

## A    USE OF LLMS

We leverage an LLM as a writing assistant to refine wording and polish manuscripts, ensuring more precise expression and ultimately enhancing the readability of the text.

## B    SYMBOLS USED IN THE TEXT

Table 5: LLMBoost Symbols

| | |
|---|---|
| $v$ | Vocabulary dimension index |
| $A_i$ | The sampling space of the model $m_i$ |
| $\mathcal{A}_i$ | The sampling space of the ensemble model $\mathcal{M}_i$ |
| $\mathcal{D}$ | Training dataset |
| $d_k$ | Dimension of key matrix |
| $e_{t,v}^{(i)}$ | Residual of $\mathcal{M}_i$ at vocab dimension $v$, t-th token |
| $\mathbb{E}$ | Expectation operator |
| $\mathcal{M}_i$ | Ensemble model at stage $i$ |
| $g$ | Global shared pool |
| $h_{l,t}^i$ | Hidden state of $i$-th model, layer $l$, step $t$ |
| $g_{t,v}^{(i)}$ | Linear term of the model $m_i$ in Taylor expansion of softmax |
| $K_{l,t}^i$ | Key matrix of $i$-th model, layer $l$, step $t$ |
| $\mathcal{L}_i$ | Loss of $i$-th model |
| $\lambda_i$ | Scaling coefficient for logits fusion |
| $M_i$ | Initial untuned base model |
| $\tilde{M}_i$ | $i$-th model after LoRA fine-tuning |
| $\mathcal{M}$ | Set of base models |
| MSE | Total Mean squared error |
| $p_t^{(i)}$ | Probability distribution of $i$-th model, t-th token |
| $P_t^{(i)}$ | Probability distribution of $\mathcal{M}_i$, t-th token |
| $Q^i$ | Query matrix of $i$-th model |
| $\mathcal{V}$ | Vocabulary set |
| $W_Q, W_K, W_V$ | Projection weight matrices |
| $x$ | Input sequence |
| $y^*$ | Ground-truth sequence |
| $\hat{y}_t$ | Token generated at step $t$ |
| $z_t^{(i)}$ | Raw logits of $i$-th model, t-th token |
| $\hat{z}_t^{(i)}$ | Accumulated logits for ensemble $\mathcal{M}_i$ at t-th token |
| EOS | End-of-sentence token |
| $\phi$ | LoRA configuration |
| $\theta_i$ | Trainable parameters of $i$-th model |
| $\Theta_{i-1}$ | Parameters of the ensemble $\mathcal{M}_{i-1}$ |
| $X_\tau$ | the input sequence |
| $Y_\tau$ | the corresponding target sequence |

## C    BROADER IMPACT AND FUTURE APPLICATIONS & LIMITATIONS

**Broader Impact and Future Applications** LLMBOOST has practical value for real-world LLM deployment. For multi-GPU tasks, it enables distributed ensemble training/inference—using layer-

wise parallelism and lightweight LoRA to boost model performance while reducing the computational overhead of full-model replication. For AI agents, it integrates directly into existing systems without modifying the agents themselves, helping them improve the reliability of cloud tasks such as tool scheduling and batch data verification via its error-correction ability, without disrupting workflows. This balances performance and deployment simplicity, fitting scenarios where both matter.

**Limitations** Despite its effectiveness, LLMBOOST currently has a key cross-model compatibility limitation: it relies on consistent structural design and hidden state formats of ensemble sub-models, restricting it to homogeneous families like LLaMA-3 or Qwen-2.5 series. For heterogeneous ensembles combining models such as LLaMA-3 and Qwen-2.5, discrepancies in hidden state dimensions, feature encoding logics, and layer-wise designs disable the residual cross-attention mechanism. This disruption impairs both the chained training paradigm—where subsequent models lack reliable error-correction signals—and the near-parallel inference paradigm—where cross-model hidden state transmission fails due to structural mismatches—ultimately making LLMBOOST inapplicable to such setups. This limitation points to future work: developing a cross-model adaptation module to bridge structural and representational gaps via targeted design strategies. Such a module would enable LLMBOOST to leverage the complementary strengths of diverse model families and expand its applicability.

## D DATASETS & BASELINES

### D.1 DATASETS INTRODUCTION

Our evaluation covers two primary task types—commonsense reasoning and arithmetic reasoning—and further extends to a domain-specific cloud service dataset (TCD).

**Commonsense Reasoning Datasets** We select six open-ended multiple-choice QA tasks to assess commonsense reasoning ability:

- **PIQA** (Bisk et al., 2020): A dataset for physical interaction commonsense reasoning, where models select the more reasonable solution from 2 options for daily physical tasks.

- **WinoGrande (WinoG.)** (Sakaguchi et al., 2021): A coreference resolution dataset with sentences containing ambiguous pronouns, requiring models to identify the correct referent via commonsense.

- **HellaSwag (HellaS.)** (Zellers et al., 2019): A scenario continuation dataset, where models select the most logically consistent ending from 4 options for a given partial daily scenario.

- **BoolQ** (Clark et al., 2019): A fact-based commonsense dataset with yes/no questions derived from web texts, requiring models to output "True" or "False" based on commonsense and text information.

- **SIQA** (Sap et al., 2019): A social commonsense dataset presenting daily social scenarios, where models choose the appropriate response from 2 options to assess social etiquette understanding.

- **Openbook QA (OBQA)** (Mihaylov et al., 2018): A structured commonsense dataset linked to a "open book" of basic scientific facts, with 4-option multiple-choice questions requiring models to apply the book's facts to answer.

**Arithmetic Reasoning Datasets** We use four open-ended math problem-solving datasets spanning multiple mathematical domains to evaluate arithmetic reasoning ability:

- **AQuA** (Ling et al., 2017): An algebraic reasoning dataset with middle-school algebra problems, requiring models to construct equations and output numerical answers.

- **GSM8K** (Cobbe et al., 2021): A multi-step arithmetic dataset with elementary math word problems, often requiring step-by-step reasoning and numerical answers.

- **MAWPS** (Koncel-Kedziorski et al., 2016): A diverse arithmetic dataset with integer, decimal, and fraction word problems, where models output numerical answers.

- **SVAMP** (Patel et al., 2021): A single-variable arithmetic dataset with word problems convertible to single-variable equations, requiring models to output numerical answers.

In both commonsense and arithmetic reasoning experiments, we follow the setup of Hu et al. (2023). For commonsense reasoning, models are fine-tuned on *Commonsense170K*, built from six benchmark training sets, and evaluated on their official test sets. For arithmetic reasoning, models are fine-tuned on *Math10K*, derived from four benchmarks, and evaluated likewise. All commonsense instances use zero-shot prompts in both fine-tuning and evaluation, while arithmetic data follow the preprocessing protocol of Wu et al. (2024) to prevent data leakage.

**Toolchain Dataset(TCD)** A domain-specific dataset for cloud service tool scheduling, derived from a specific laboratory's real user interaction logs. Each sample includes a predefined tool library, a user inquiry, and requires models to select the most appropriate tools from the library to address the inquiry.

### D.2 BASELINES

We further analyze the improvements of LLMBOOST over representative ensemble methods:

- **Vote** (Li et al., 2024a): The vote-based method for large model ensembling aggregates predictions from multiple individual models by having them vote on the final output.It leverages the complementary strengths of different models to reduce individual model biases and improve overall prediction robustness.

- **UNITE**(Yao et al., 2024): UNITE is a novel LLM ensembling method that efficiently combines models by focusing on the union of top-k tokens from each model, avoiding full vocabulary alignment and reducing computational overhead.It incorporates a model selection strategy identifying compatible models, leveraging their complementary advantages to enhance performance across multiple benchmarks.

- **T-copilot**(Zou et al., 2025): T-copilot is a method that adapts large language models to downstream tasks by retaining the model's own learning signals, using a Pilot-Copilot framework with joint training and fused inference.It tracks the Pilot model's errors via a Mistake Log, which the Copilot uses to rectify the Pilot's logits for better generation.

## E NEAR-PARALLEL DECODING PSUDO ALGORITHM

The pseudo code in Algorithm 2 illustrates the overall workflow of our near-parallel decoding strategy. The key idea is to allow multiple models to decode tokens in parallel while maintaining synchronization through a lightweight communication mechanism. By progressively sharing intermediate hidden states, the framework achieves both efficiency and consistency in multi-model inference.

Specifically, the global shared pool $g$ is implemented as a thread-safe queue that stores hidden states produced by each model at every layer. Once a model finishes computing a hidden state $h_{l,t}^{(i)}$, it immediately pushes the result into $g$, where it can be consumed by downstream models. This ensures that information flows smoothly across different models without requiring full-sequence synchronization.

In practice, we employ multi-threading for parallel execution, where each auxiliary model $\tilde{m}_i$ runs in a separate thread. To guarantee correctness, a lightweight lock mechanism is introduced so that models can safely wait for the availability of required hidden states in $g$ before continuing computation. This design provides fine-grained synchronization with minimal overhead, enabling near-parallel decoding while avoiding race conditions.

---

**Algorithm 2:** Near-Parallel Decoding Psudo Algorithm

---

**Input:** Model list $\tilde{\mathcal{M}} = \{\tilde{m}_0(\cdot; \theta_0), \tilde{m}_1(\cdot; \theta_1), \ldots, \tilde{m}_n(\cdot; \theta_n)\}$; input $x$; max Tokens $T$; global
  shared pool $g$; hidden states $h_{l,t}^{(i)}$ of the $i$-th model at layer $l$ for token $t$.

**Output:** Final output $\{\hat{y}_0 \ldots \hat{y}_t\}$

1 Initialize shared pool $g \leftarrow \emptyset$;
2 Run $\tilde{m}_0$ in main process, others $\tilde{m}_i$ $(i \geq 1)$ in parallel threads;
3 **for** $t = 1$ **to** $T$ **do**
4    **parfor** $i = 1$ **to** $n$ **do**
5       **for** $l = 1$ **to** $L$ **do**
6          Each model $\tilde{m}_i$ waits for $h_{l,t}^{(i-1)}$ in $g$;
7          Compute hidden state $h_{l,t}^{(i)}$ at layer $l$ with available inputs;
8          Immediately push $h_{l,t}^{(i)}$ into $g$ for downstream use;
9       **end**
10    **end**
11    Each model $\tilde{m}_i$ produces t-th logits $z_t^{(i)}$;
12    Fuse logits $z_t^0$ according to Eq. (5);
13    Decode token $\hat{y}_t \leftarrow \text{Argmax}(\tilde{z}_t^0)$ ;
14    **if** $\hat{y}_t = EOS$ **then**
15       break
16    **end**
17 **end**
18 **return** $\{\hat{y}_0 \ldots \hat{y}_t\}$

---

# F FINE-TUNING DATASET TEMPLATE

Table 6: Examples of dataset templates used in LLMBoost.

| Task Type | Template |
|---|---|
| Commonsense Reasoning | **Input:** Please choose the correct solution to the question: To fight Ivan Drago in Rocky for sega master system. Solution1: Drago isn't in this game because it was released before Rocky IV. Solution2: You have to defeat Apollo Creed and Clubber Lang first. Answer format: solution1/solution2

**Answer:** the correct answer is solution2. |
| | **Input:** Please choose the correct answer to fill in the blank to complete the given sentence: Sarah was a much better surgeon than Maria so () always got the harder cases. Option1: Sarah Option2: Maria Answer format: option1/option2

**Answer:** the correct answer is option1. |
| | **Input:** Please answer the following question with true or false, question: is there a difference between hydroxyzine hcl and hydroxyzine pam? Answer format: true/false

**Answer:** the correct answer is true. |
| Arithmetic Reasoning | **Input:** In a school there are 569 girls and 236 boys. How many more girls than boys does the school have?

**Answer:** There are 569 girls and 236 boys. The difference between them is 569 - 236 = 333. The school has 333 more girls than boys. The answer is 333. |
| | **Input:** A study reported that in a random sampling of 100 women over the age of 35 , 8 of the women were married 2 or more times . Based on the study results , how many woman in a group of 5000 women over the age of 35 would likely be married 2 or more times ?

**Answer:** x=(8.0/100.0)*5000.0 |
| | **Input:** Twenty kids went out on a school trip. In one of the several activities they participated in, they were divided into two equal groups of girls and boys and then went out for a scavenger hunt of seashells. The boys went out for the search and brought back 60 shells each. If the girls brought an amount equal to the number of shells brought by the boys plus four times as many seashells as the boys, how many seashells were brought by each girl?

**Answer:** The boys brought back 60 shells each. That means the girls must have brought an amount equal to the number of shells brought by the boys plus four times as many seashells as the boys. That means the girls must have brought 60 + 4 * 60 = 300 shells each. The answer is 300. |
| TCD | **Input:** cloud agent can use following tools: (1)<name>QueryCloudServerCount</name><describe>can query the number of cloud server under the user's account</describe> (2)<name>QueryUserQuota</name><describe>Query the global upper limit of the number of Elastic Public IPs (EIPs), as well as the upper limits in a specified resource pool, including the upper limit of Virtual Machines (VMs), the upper limit of Virtual Private Clouds (VPCs), the upper limit of the total number of storage disks, and the upper limit of the total capacity (in GB) of storage disks.</describe> (3)<name>QueryCloudServerInformation</name><describe>Query cloud server specifications information by <param>user_id</param>and <param>cloud_server_id</param></describe> (4)<name>QueryCloudResourceList</name><describe>Query the names of all cloud resources by <param>user_id</param></describe> When cloud agent receives an inquiry: Can I create 2 more cloud hosts? Which tool is Cloud Agent most likely to call? Answer Choices: (A) QueryCloudServerCount and QueryUserQuota (B) QueryUserQuota (C) QueryCloudServerCount and QueryCloudServerInformation (D) QueryCloudServerInformation and QueryCloudResourceList

**Answer:** A. |

In Table 6, we provide examples of data instances for each task mentioned above during model fine-tuning. All experiments are conducted in the zero-shot setting to better facilitate model-wise evaluation using accuracy.

# G INFERENCE TIME EFFICIENCY ANALYSIS

We analyze the inference latency under different execution strategies.

**Sequential execution.**    In the naive setting, each model $\tilde{M}_i$ must finish all $L$ layers before the next model starts, with per-layer compute cost $c$ . The total latency is

$$T_{\text{sequential}} \;\approx\; (n+1) \cdot L \cdot c. \tag{14}$$

**Parallel execution with inter-model dependency.**    We consider $k$ models, each with $l$ layers, executed on $g$ GPUs in a round-robin assignment. Let the execution of one layer on a dedicated GPU take time $c$. Due to inter-model dependencies, the $i$-th layer of model $m$ can only start after the $i$-th layer of model $m-1$ and the $(i-1)$-th layer of model $m$ have completed. Hence, the computation graph forms a $k \times l$ grid with precedence constraints along both the horizontal and vertical directions.

Define an anti-diagonal index $s = m + i$, so that all tasks $\{(m,i) : m+i = s\}$ can be executed in parallel once their predecessors complete. The number of tasks on the $s$-th anti-diagonal is

$$d_s \;=\; \min(k,\, s-1) - \max(1,\, s-l) + 1. \tag{15}$$

These values increase from 1 to $w = \min(k,l)$, stay constant at $w$ for $u - w + 1$ steps where $u = \max(k,l)$, and then decrease symmetrically back to 1. Geometrically, the execution order thus corresponds to a *parallelogram*, which can be decomposed into two symmetric triangular parts and a central rectangular part.

On the $s$-th anti-diagonal, round-robin assignment ensures that tasks are distributed across GPUs as evenly as possible. Therefore, the maximum number of tasks assigned to a single GPU is exactly $\lceil d_s/g \rceil$, and the completion time of this anti-diagonal is

$$T_s \;=\; c \cdot \left\lceil \frac{d_s}{g} \right\rceil. \tag{16}$$

Summing over all anti-diagonals, the total execution time is

$$T_{\text{parallel}} \;=\; c \sum_{s=2}^{k+l} \left\lceil \frac{d_s}{g} \right\rceil. \tag{17}$$

By decomposing the parallelogram into two triangles and one rectangle, we obtain the closed-form expression

$$T_{\text{parallel}} \;=\; 2c \sum_{t=1}^{w-1} \left\lceil \frac{t}{g} \right\rceil \;+\; (u-w+1)c \left\lceil \frac{w}{g} \right\rceil + \delta, \tag{18}$$

where $w = \min(k,l)$, $u = \max(k,l)$ and $\delta$ is communication overhead.

**Speedup over sequential execution.**    The theoretical speedup compared to the naive sequential strategy is

$$\text{Speedup} \;=\; \frac{T_{\text{sequential}}}{T_{\text{parallel}}} \;=\; \frac{(n+1) \cdot L}{2 \sum\limits_{t=1}^{w-1} \left\lceil \frac{t}{g} \right\rceil + (u-w+1) \left\lceil \frac{w}{g} \right\rceil + \delta}. \tag{19}$$

This expression clearly shows the achievable acceleration: (i) if $g = 1$, the speedup is 1, as expected; (ii) if $g \geq w$, the parallel latency reduces to $c(k+l-1)$, and the speedup approaches $\frac{kl}{k+l-1}$, which scales almost linearly in $\min(k,l)$; (iii) for intermediate $g$, the speedup smoothly interpolates between these two extremes, depending on how many anti-diagonal tasks can be executed simultaneously.

# H   ADDITIONAL EXPERIMENTS

## H.1   IMPLEMENTATION DETAILS.

Tables 7 and 8 summarize the main training details for reproducibility. In our model, the fusion coefficient of cross-model attention and the coefficient for error suppression are both set to 0.1. All reported results can be reproduced with random seeds 1, 2, and 3.

Table 7: Unified hyperparameter configuration of LLMBOOST for LLaMA-3 and Qwen-2.5 series models on **Commonsense Reasoning** tasks.

| Hyperparameters | Llama Models | | Qwen Models | |
|---|---|---|---|---|
| | LLaMA-3.2-3B | LLaMA-3.1-8B | Qwen-2.5-3B | Qwen-2.5-7B |
| *Fine-tuning Configurations* | | | | |
| Epochs | 3 | 3 | 3 | 3 |
| Batch Size | 16 | 16 | 16 | 16 |
| Micro Batch Size | 4 | 4 | 4 | 4 |
| Cut Off Length | 256 | 256 | 256 | 256 |
| Maximum Learning Rate | $3e^{-4}$ | $3e^{-4}$ | $3e^{-4}$ | $3e^{-4}$ |
| Learning Rate Scheduler | Cosine | Cosine | Cosine | Cosine |
| Optimizer | AdamW | AdamW | AdamW | AdamW |
| Warmup Steps | 200 | 200 | 200 | 200 |
| Weight Decay | 0.00 | 0.00 | 0.00 | 0.00 |
| *LoRA Configurations* | | | | |
| Rank $r$ | 32 | 32 | 32 | 32 |
| LoRA Alpha | 64 | 64 | 64 | 64 |
| LoRA Dropout | 0.05 | 0.05 | 0.05 | 0.05 |
| *Inference Configurations* | | | | |
| Temperature | 0.1 | | | |
| Top p | 0.95 | | | |
| Top k | 40 | | | |
| Num Beams | 4 | | | |
| Maximum New Tokens | 64 | | | |

Table 8: Unified hyperparameter configuration of LLMBOOST for LLaMA-3 and Qwen-2.5 series models on **Arithmetic Reasoning** tasks.

| Hyperparameters | Llama Models | | Qwen Models | |
|---|---|---|---|---|
| | LLaMA-3.2-3B | LLaMA-3.1-8B | Qwen-2.5-3B | Qwen-2.5-7B |
| *Fine-tuning Configurations* | | | | |
| Epochs | 3 | 3 | 3 | 3 |
| Batch Size | 16 | 16 | 16 | 16 |
| Micro Batch Size | 4 | 4 | 4 | 4 |
| Cut Off Length | 256 | 256 | 256 | 256 |
| Maximum Learning Rate | $3e^{-4}$ | $3e^{-4}$ | $3e^{-4}$ | $3e^{-4}$ |
| Learning Rate Scheduler | Cosine | Cosine | Cosine | Cosine |
| Optimizer | AdamW | AdamW | AdamW | AdamW |
| Warmup Steps | 100 | 100 | 100 | 100 |
| Weight Decay | 0.00 | 0.00 | 0.00 | 0.00 |
| *LoRA Configurations* | | | | |
| Rank $r$ | 32 | 32 | 32 | 32 |
| LoRA Alpha | 64 | 64 | 64 | 64 |
| LoRA Dropout | 0.05 | 0.05 | 0.05 | 0.05 |
| *Inference Configurations* | | | | |
| Temperature | 0.1 | | | |
| Top p | 0.95 | | | |
| Top k | 40 | | | |
| Num Beams | 4 | | | |
| Maximum New Tokens | 256 | | | |

## H.2 ABLATION STUDY

To better understand the contribution of each component in our method, we conduct ablation experiments to verify the effectiveness of different modules. Unless otherwise stated, all experiments are performed under the same settings as in the main results.

We first investigate the effectiveness of the key components in our method. Specifically, we analyze the impact of the Error-Suppression Objective(E.S.O.) and cross-model attention(C.A.). We compare LLMBOOST with its ablated variants to understand the contribution of each component.

Table 9: Ablation study of key components in LLMBOOST, evaluated on mathematical reasoning benchmarks (SVAMP, MAWPS). "wo E.S.O." removes the Error-Suppression Objective; "wo C.A." disables Cross-Model Attention; "wo $top\text{-}k$" ablates the Key-Token Backward Pass (with $k = 2$). LLMBOOST consistently outperforms all variants, validating the contribution of each module.

| MODEL TYPE | MAWPS | SVAMP | Avg. |
|---|---|---|---|
| **Qwen-2.5-7B as base model** | | | |
| LLMBOOST ($2 \times 7B$) | 88.8 | 94.1 | 91.5 |
| wo E.S.O. | 87.3 | 93.3 | 90.3 |
| wo C.A. | 87.4 | 92.4 | 89.9 |
| wo $top\text{-}k$ | 87.2 | 93.3 | 90.3 |

Table 10: Sensitivity analysis of the weighting coefficient(s) $\lambda$ in LLMBOOST. Performance (%) is evaluated across arithmetic reasoning benchmarks (AQuA, MAWPS, SVAMP) under matched parameter scales. The upper block explores $\lambda_1 \in [0.1, 1.0]$ for the LLMBOOST ($2 \times 8B$), while the lower block studies combinations of $\lambda_1, \lambda_2$ for the LLMBOOST ($3 \times 8B$).

| Params | $\lambda$ | AQuA | MAWPS | SVAMP | Avg. |
|---|---|---|---|---|---|
| **Llama-3.1-8B as base model** | | | | | |
| | 0.1 | 45.7 | 89.4 | 78.9 | 71.3 |
| | 0.2 | 46.4 | 90.1 | 80.0 | 72.2 |
| | 0.3 | 46.8 | 90.1 | 80.1 | **72.3** |
| | 0.4 | 43.9 | 90.1 | 80.1 | 71.4 |
| $2 \times 8B$ | 0.5 | 46.3 | 89.3 | 79.8 | 71.8 |
| | 0.6 | 46.1 | 90.3 | 79.6 | 72.0 |
| | 0.7 | 43.9 | 89.7 | 80.0 | 71.2 |
| | 0.8 | 45.3 | 90.7 | 79.9 | 72.0 |
| | 0.9 | 45.7 | 89.9 | 79.5 | 71.7 |
| | 1.0 | 45.6 | 89.8 | 79.2 | 71.5 |
| | 0.3, 0.1 | 46.9 | 90.7 | 80.4 | 72.7 |
| $3 \times 8B$ | 0.3, 0.2 | 46.6 | 92.1 | 80.8 | 73.2 |
| | 0.3, 0.3 | 47.6 | 92.0 | 81.3 | **73.6** |

## H.3 EFFECT OF $\lambda$ CHOICE

In addition to ablation studies, we conduct a detailed sensitivity analysis of the weighting coefficient(s) $\lambda$, which control the balance between different loss components in LLMBOOST. As shown in Table 10, for the two-module ($2 \times 8B$) configuration, performance peaks when $\lambda_1 = 0.3$, yielding an average accuracy of 72.3%, while overly large or small values lead to suboptimal results. For the three-module ($3 \times 8B$) ensemble, the best performance of 73.6% is obtained with ($\lambda_1 = 0.3, \lambda_2 = 0.3$), suggesting that balanced weighting across modules is essential. These findings highlight that careful tuning of $\lambda$ is critical for maximizing the benefit of multi-model integration.

## H.4 TOP-K SELECTION

We further examine how different top-$k$ settings affect decoding performance, with results summarized in Table 11. Performance remains largely stable across $k$ values, showing only minor variations. The best average accuracy **91.2%** is achieved at $k = 2$, slightly outperforming other choices. This suggests that a small candidate set is sufficient to strike a balance between exploration and output stability during decoding. Notably, increasing $k$ beyond 3 yields diminishing returns: performance plateaus or even dips slightly, indicating that larger candidate pools offer little added value for reasoning tasks like SVAMP and MAWPS.

Table 11: Performance comparison (%) under different top-$k$ selections on logits.

| Params | k | SVAMP | MAWPS | Avg. |
|---|---|---|---|---|
| **Qwen-2.5-7B as base model** | | | | |
| | 1 | 87.9 | 92.9 | 90.4 |
| | 2 | 88.8 | 94.1 | **91.2** |
| | 3 | 88.3 | 93.3 | 90.8 |
| $2 \times 7B$ | 4 | 87.3 | 93.4 | 90.4 |
| | 5 | 87.4 | 93.3 | 90.4 |
| | 6 | 87.1 | 93.3 | 90.2 |
| | 7 | 87.2 | 93.3 | 90.3 |
| | 8 | 87.2 | 93.3 | 90.3 |

Table 12: Performance comparison (%) under different *sparsity of cross-model connections* settings, i.e., varying the interval of inter-layer interactions.

| Params | $\phi$ | SVAMP | MAWPS | Avg. |
|---|---|---|---|---|
| **Qwen-2.5-7B as base model** | | | | |
| | 1 | 88.4 | 93.3 | 90.9 |
| $2 \times 7B$ | 2 | 88.8 | 94.1 | 91.5 |
| | 3 | 88.1 | 93.3 | 90.7 |
| | 4 | 86.8 | 92.9 | 89.9 |

Table 13: Performance comparison (%) under different $\alpha$ selections.

| Params | $\alpha$ | SVAMP | MAWPS | Avg. |
|---|---|---|---|---|
| **Qwen-2.5-7B as base model** | | | | |
| | 1.00 | 87.0 | 93.7 | 90.4 |
| | 0.99 | 87.0 | 93.7 | 90.4 |
| $2 \times 7B$ | 0.95 | 88.2 | 93.7 | 91.1 |
| | 0.90 | **88.8** | **94.1** | **91.5** |
| | 0.80 | 87.9 | 92.9 | 90.4 |
| | 0.50 | 87.9 | 92.4 | 90.2 |

Table 14: Performance comparison (%) under different $\beta$ selections.

| Params | $\beta$ | SVAMP | MAWPS | Avg. |
|---|---|---|---|---|
| **Qwen-2.5-7B as base model** | | | | |
| | 0.01 | 87.0 | 92.4 | 89.7 |
| | 0.05 | 88.1 | 93.7 | 90.9 |
| | 0.10 | 88.8 | 92.4 | **91.5** |
| $2 \times 7B$ | 0.20 | 89.0 | 92.4 | 90.7 |
| | 0.50 | 88.5 | 92.4 | 90.5 |

## H.5 Effect of Cross-Model Connection Sparsity

We conduct experiments to test how the sparsity of cross-model connections controlled by $\phi$ affects performance, using Qwen-2.5-7B as the base model in a $2 \times 7B$ ensemble. We test $\phi = 1$ to 4, fix other hyperparameters and evaluate on SVAMP and MAWPS. Results show $\phi = 2$ achieves the best performance, with higher accuracy on both datasets than other $\phi$ values. $\phi = 1$ frequent cross-model fusion causes inefficient computation without extra gains, while $\phi \geq 3$ over-sparse fusion loses valuable intermediate information, leading to reduced performance.

## H.6 $\alpha \& \beta$ Selection

In the loss function, the choices of $\alpha$ and $\beta$ form an $n^2$ search space, which makes exhaustive exploration computationally infeasible. To reduce the search cost, we first fix $\alpha = 1.0$ and sweep

Table 15: Efficiency Comparison of Different Models (Time/Memory Metrics)

| Model | End-to-End (s) | Token Cost (s) | Peak Train Mem (%) | Peak Infer Mem (%) |
|---|---|---|---|---|
| Qwen14B | 17.40 | 0.098 | 56.74% | 36.97% |
| LLMBoost 7B+7B (near-par, 2 GPUs) | 13.75 | 0.075 | 33.18% | 20.72% |
| LLMBoost 7B+7B (near-par, 1 GPU) | 14.87 | 0.081 | 54.11% | 39.75% |
| LLMBoost 7B+7B+7B (near-par, 3 GPUs) | 18.88 | 0.103 | 33.18% | 20.13% |
| LLMBoost 7B+7B+7B (near-par, 1 GPU) | 21.67 | 0.118 | 73.42% | 58.48% |
| LLMBoost 7B+7B (seq) | 22.53 | 0.125 | 60.29% | 38.72% |
| VOTE (7B+7B) | 22.61 | 0.127 | 53.39% | 39.14% |
| T-Copilot (7B+7B) | 17.21 | 0.097 | 64.41% | 42.05% |
| UNITE (7B+7B) | 22.66 | 0.128 | 53.39% | 39.12% |

across different $\beta$ values. After identifying the best-performing $\beta$, we then fix this value and sweep $\alpha$. The results summarized in Table 13 and Table 14 show that LLMBOOST achieves its best overall performance when both hyperparameters are set to $\alpha = 0.90$ and $\beta = 0.10$. These observations verify that our two-stage search strategy is effective and that moderate values of $\alpha$ and $\beta$ provide a good balance between suppression strength and optimization stability.

### H.7 DETAILED EFFICIENCY RESULTS

In this appendix, we provide a detailed comparison between single-model baselines (Qwen2.5-14B, Qwen2.5-7B), traditional ensemble methods (Vote, TCopilot, Unite), and our proposed LLMBOOST under different execution strategies. Here, Seq denotes sequential execution, where multiple 7B sub-models are evaluated one after another on the same computation path; Par denotes parallel execution, where sub-models are evaluated concurrently; and GPU=1 indicates that all sub-models are colocated on a single GPU, whereas GPU > 1 distributes different sub-models across multiple GPUs.

In terms of end-to-end latency, Qwen2.5-14B serves as the single-model baseline (17.40 s, 0.098 s/token). Notably, LLMBOOST (Qwen2.5-7B×2 Par, GPU=2) achieves 13.75 s end-to-end latency and 0.075 s/token, delivering a two-model ensemble that is not only faster than Qwen2.5-14B but also significantly more efficient than traditional ensembles such as Vote (22.61 s, 0.127 s/token). This demonstrates that under genuine multi-GPU parallelism, the Par strategy can reap ensemble gains with manageable latency overhead while outperforming the single 14B model. By contrast, LLMBOOST (Qwen2.5-7B×2 Seq), which performs linear execution on a single device, increases end-to-end latency to 22.53 s—closely matching the behavior of Vote and Unite in the 22–23 s range—and exhibits the expected near-linear growth of runtime with respect to the number of sub-models. Similarly, LLMBOOST (Qwen2.5-7B×3 Par, GPU=1) reaches 21.67 s, illustrating that when all models reside on a single GPU, "parallel" execution is effectively limited by the same hardware budget and cannot realize true multi-stream speedup.

From the memory perspective, Qwen2.5-14B (the single-model baseline) requires 56.74% peak training memory and 36.97% peak inference memory. Under multi-GPU parallel settings, LLM-BOOST-7B×2 (Par, GPU=2) and LLMBOOST (Qwen2.5-7B×3 Par, GPU=3) both maintain training peak memory around 33.18%, with inference peaks of 20.72% and 20.13%, respectively—far lower than the single Qwen2.5-14B baseline. This indicates that distributing sub-models across GPUs allows Par execution to "spread" the memory footprint rather than stacking it on a single device. In contrast, single-GPU multi-model configurations (e.g., LLMBOOST (Qwen2.5-7B×3 Par, GPU=1) with 73.42% training and 58.48% inference peak memory, or LLMBOOST (Qwen2.5-7B×2 Seq) with 60.29% training) and traditional ensembles (Vote at 53.39%, T-Copilot at 64.41% training peak) exhibit substantial cumulative memory overhead as the number of models increases.

Overall, these observations support three key conclusions: (1) Seq (sequential) execution on a single GPU leads to almost linear growth in both latency and memory usage with respect to the ensemble size; (2) Par (parallel) execution over multiple GPUs achieves faster latency than the single Qwen2.5-14B model while keeping per-device memory consumption significantly lower; and (3) under the same hardware budget, LLMBOOST strikes a strictly better trade-off between end-to-end

Table 16: Performance of different models on 4 core GLUE subtasks (%)

| Model | CoLA | SST-2 | STS-B | MNLI | Avg. |
|---|---|---|---|---|---|
| Qwen2.5-14B | 68.31 | 95.78 | 91.93 | 90.77 | 86.70 |
| tcopilot (Qwen2.5-7B × 2) | 68.84 | 67.43 | 91.54 | 90.40 | 79.55 |
| vote (Qwen2.5-7B × 2) | 69.03 | 95.99 | 92.10 | 90.64 | 86.94 |
| unite (Qwen2.5-7B × 2) | 68.93 | 95.99 | 91.67 | 90.58 | 86.79 |
| LLMBoost (Qwen2.5-7B × 2) | 69.13 | 96.22 | 92.08 | 90.96 | 87.10 |

latency, per-token efficiency, and memory consumption than Qwen2.5-14B and existing ensemble baselines, making it a more practical option for real-world multi-GPU deployment.

### H.8 EVALUATE ON GLUE BENCHMARKS

This section supplements evaluations on core subtasks of the GLUE benchmark to verify the model's general language understanding capability and cross-task transferability.

We select four representative GLUE subtasks and their corresponding metrics to reflect model performance across diverse language understanding scenarios. In terms of data selection, we prioritize MNLI (the most representative task in GLUE) and further evaluate datasets with sample sizes ranging from 5k to 100k, balancing computational efficiency and the comprehensiveness of evaluation results.

Table 16 presents the performance of 5 models on the four subtasks, covering grammatical acceptability judgment (CoLA, evaluated by MCC), sentiment analysis (SST-2, evaluated by Acc), correlation prediction (STS-B, evaluated by Pearson), and natural language inference (MNLI, evaluated by Acc).

As shown in Table 16, LLMBOOST achieves the best performance on 3 subtasks (CoLA, SST-2, MNLI) and ranks 2nd on STS-B. It consistently outperforms the single-model baseline and other ensemble baselines, confirming the effectiveness of the proposed multi-model collaboration mechanism and its transferability across diverse NLP tasks.

### H.9 DETAILED DESCRIPTION OF E2E TESTING

The end-to-end toolchain scheduling tests, built on a specific laboratory's dedicated test sets, aim to evaluate the capability of large language model to accurately execute tool-call tasks in a specific laboratory's internal workflows.

The test follows a standardized process: a user query and an expected tool call are input into the model via OpenAI's /v1/chat/completions interface, and the returned tool calls are then compared with the expected ones to determine whether a test case passes.

Each test input includes a user query and an expected_tool_call, while the output is the model's response, containing the actual tool_calls and a finish reason.

When using two Qwen-2.5-7B models, the overall pass rate of 94 test cases reaches 24% (23/94); when using one Qwen-2.5-7B model, the pass rate drops to 19% (18/94)—demonstrating that multi-model setups outperform single-model setups in tool-call accuracy.

### H.10 CROSS-TASK PERFORMANCE AND EFFICIENCY: LLMBOOST VS. 70B SINGLE MODELS

LLMBoost offers a practical alternative when 70B models are infeasible and single 8B models are insufficient, balancing competitive performance and high efficiency via lightweight fine-tuning.

In commonsense reasoning tasks (see Table 17), LLMBoost (Llama3.1-8B × 3) closely approximates Llama3.1-70B's capabilities: it scores 95.2 on HellaSwag (only 1.7 points lower than the 70B model) and achieves an average accuracy of 84.5—neatly filling the gap between the single 8B variant (80.0) and the 70B model (89.2). Even on tasks like SIQA, the performance gap remains narrow

Table 17: Model Performance on Commonsense Reasoning Tasks

| Model | Params | Latency | PIQA | WinoG | HellaS. | Boolq | SIQA | OBQA | Avg |
|---|---|---|---|---|---|---|---|---|---|
| Llama3.1-70B | 70B | 0.254 | 93.2 | 93.1 | 96.9 | 76.5 | 82.4 | 92.0 | 89.2 |
| LLMBoost (8B×3) | 24B | 0.064 | 87.4 | 86.3 | 95.2 | 71.7 | 80.7 | 84.8 | 84.5 |
| LLMBoost (8B×1) | 8B | 0.038 | 83.3 | 81.8 | 90.6 | 69.5 | 75.8 | 79.0 | 80.0 |

(80.7 vs. 70B's 82.4), while it maintains a low latency of 0.064 s/token—far outperforming the 70B model's 0.254 s/token in inference speed.

Table 18: Model Performance and Efficiency Comparison

| Model | Params | Latency | AQUA | GSM8K | MAWPS | SVAMP | Avg |
|---|---|---|---|---|---|---|---|
| Llama3.1-70B | 70B | 0.254 | 59.1 | 77.4 | 91.6 | 82.6 | 77.7 |
| LLMBoost (8B × 3) | 24B | 0.064 | 47.6 | 68.5 | 92.0 | 81.3 | 72.4 |
| LLMBoost (8B × 1) | 8B | 0.038 | 42.3 | 63.7 | 89.5 | 77.4 | 68.2 |

In arithmetic reasoning tasks (Table 18), LLMBoost demonstrates strong performance with its ensemble design: LLMBoost (Llama3.1-8B × 3) achieves an average score of 72.4, and surpasses the 70B model on the MAWPS benchmark (92.0 vs. 91.6), showcasing targeted excellence in numerical reasoning. Even the single 8B sub-model variant (LLMBoost (Llama3.1-8B × 1)) delivers a solid average accuracy of 68.2 while maintaining extreme inference efficiency. In terms of speed, the single 8B variant runs at only 0.038 s/token, and the 3-model ensemble still requires just 0.064 s/token—both drastically outpacing the 70B model's 0.254 s/token latency.

Overall, LLMBoost narrows the performance gap with 70B models across both commonsense and arithmetic tasks, while delivering drastically lower latency. This makes it a resource-efficient solution for scenarios where large-scale models are impractical to deploy.

# I PROOFS OF THE MAIN THEORETICAL RESULTS

## I.1 FORMAL SETUP FOR PROOFS

This section provides detailed proofs for the theorems presented in the main body. The analysis is conducted for an arbitrary decoding step $t$ during inference. For notational brevity, we adopt the following conventions throughout the proofs.

For the analysis at a given decoding step $t$, let $m_i$ denote an individual model and $A_i$ its sampling space. The logits produced by this model at step $t$ are given by $z_t^{(i)}$. We consider an ensemble of models $\mathcal{M}_i = \{m_0, \ldots, m_i\}$, with a corresponding sampling space $\mathcal{A}_i$. The accumulated logits for this ensemble, $\hat{z}_t^{(i)}$, are defined recursively as $\hat{z}_t^{(i)} = \hat{z}_t^{(i-1)} + \lambda_i z_t^{(i)}$, starting with the base case $\hat{z}_t^{(0)} = z_t^{(0)}$. From these logits, the ensemble's predicted probability distribution is calculated as $\hat{P}_t^{(i)} = \mathrm{softmax}(\hat{z}_t^{(i)})$. Let $\hat{y}_{<t}^{(i-1)}$ be the sequence of tokens generated by the ensemble $\mathcal{M}_{i-1}$ up to step $t - 1$. The ground truth probability distribution for the token at step $t$, conditioned on an input $X_\tau$ and the generated prefix, is $p_t^*(\cdot \mid X_\tau, \hat{y}_{<t}^{(i-1)})$, which we denote by the one-hot vector $p_t^*$. The residual error of the ensemble $\mathcal{M}_{i-1}$ can thus be expressed as the difference between the true and the predicted distributions, $e_t^{(i-1)} = p_t^* - \hat{P}_t^{(i-1)}$. To quantify how a new model $m_i$ influences the ensemble's prediction, we define its effective linear contribution to the probability space as $g_t^{(i)} \triangleq J_{\mathrm{softmax}}(\hat{z}_t^{(i-1)})z_t^{(i)}$, where $J_{\mathrm{softmax}}(\hat{z}_t^{(i-1)})$ is the Jacobian matrix of the softmax function evaluated at the accumulated logits of the preceding ensemble, $\hat{z}_t^{(i-1)}$.

The expectation $\mathbb{E}[\cdot]$ is taken over all relevant sources of randomness (data distribution $\mathcal{D}$ and model training distributions $\mathcal{A}_i$). We omit the explicit conditioning on the input $X_\tau$ and the generated prefix $\hat{y}_{<t}^{(i-1)}$ for clarity, though all probabilistic quantities are implicitly dependent on this context.

## I.2 PRELIMINARY LEMMA FOR MSE CHANGE

**Lemma 1** (Asymptotic Form of Single-Dimension MSE Change). *The change in MSE for a single vocabulary dimension $v$ at step $t$, denoted $\Delta_{\mathrm{MSE},t,v} = \mathbb{E}[(e_{t,v}^{(i)})^2] - \mathbb{E}[(e_{t,v}^{(i-1)})^2]$, resulting from the addition of successor $m_i$ with a small scaling coefficient $\lambda_i$, can be expressed as:*

$$\Delta_{\mathrm{MSE},t,v} = -2\lambda_i \mathbb{E}[e_{t,v}^{(i-1)} g_{t,v}^{(i)}] + O(\lambda_i^2). \tag{20}$$

*Proof.* The new ensemble probability is $\hat{P}_t^{(i)} = \mathrm{softmax}(\hat{z}_t^{(i-1)} + \lambda_i z_t^{(i)})$. Using a first-order Taylor expansion, the new residual vector $e_t^{(i)} = p_t^* - \hat{P}_t^{(i)}$ can be written as:

$$e_t^{(i)} = p_t^* - \left(\hat{P}_t^{(i-1)} + \lambda_i J_{\mathrm{softmax}}(\hat{z}_t^{(i-1)})z_t^{(i)} + R(\lambda_i z_t^{(i)})\right) \tag{21}$$

$$= (p_t^* - \hat{P}_t^{(i-1)}) - \lambda_i g_t^{(i)} - R(\lambda_i z_t^{(i)}) \tag{22}$$

$$= e_t^{(i-1)} - \lambda_i g_t^{(i)} - R(\lambda_i z_t^{(i)}), \tag{23}$$

where the remainder vector $R(\cdot)$ satisfies $\|R(\Delta z)\|_2 = O(\|\Delta z\|_2^2)$. Let $R_{t,v}$ be the $v$-th component of $R(\lambda_i z_t^{(i)})$. The change in MSE is $\Delta_{\mathrm{MSE},t,v} = \mathbb{E}[(e_{t,v}^{(i-1)} - \lambda_i g_{t,v}^{(i)} - R_{t,v})^2] - \mathbb{E}[(e_{t,v}^{(i-1)})^2]$. Expanding the squared term and using linearity of expectation gives:

$$\Delta_{\mathrm{MSE},t,v} = -2\lambda_i \mathbb{E}[e_{t,v}^{(i-1)} g_{t,v}^{(i)}] + \lambda_i^2 \mathbb{E}[(g_{t,v}^{(i)})^2] - 2\mathbb{E}[e_{t,v}^{(i-1)} R_{t,v}] + 2\lambda_i \mathbb{E}[g_{t,v}^{(i)} R_{t,v}] + \mathbb{E}[R_{t,v}^2]. \tag{24}$$

As shown in Lemma 4, all terms involving the remainder $R_{t,v}$ are of order $O(\lambda_i^2)$ or higher. For a sufficiently small $\lambda_i > 0$, the linear term dominates, leading to the stated asymptotic form. $\square$

## I.3 PROOF OF THEOREM 1

**Theorem 2** (Restate). *Under Assumption 1, for each stage $i \geq 1$, decoding step $t$, and vocabulary dimension $v$, there exists an upper bound $\lambda_{i,t,v}^* > 0$ for the scaling coefficient. For any weight $\lambda_i \in (0, \lambda_{i,t,v}^*)$, the MSE of the new ensemble $\mathcal{M}_i$ is strictly lower than that of $\mathcal{M}_{i-1}$ for that dimension.*

*Proof.* We analyze the change in MSE, $\Delta_{\mathrm{MSE},t,v} = \mathbb{E}[(e_{t,v}^{(i)})^2] - \mathbb{E}[(e_{t,v}^{(i-1)})^2]$. Our goal is to find a $\lambda_i > 0$ such that $\Delta_{\mathrm{MSE},t,v} < 0$. By Lemma 1, the asymptotic form is:

$$\Delta_{\mathrm{MSE},t,v} = -2\lambda_i \mathbb{E}[e_{t,v}^{(i-1)} g_{t,v}^{(i)}] + O(\lambda_i^2). \tag{25}$$

For a sufficiently small $\lambda_i > 0$, the sign of $\Delta_{\mathrm{MSE},t,v}$ is determined by $-\mathbb{E}[e_{t,v}^{(i-1)} g_{t,v}^{(i)}]$. We now show this expectation is strictly positive.

By the law of total expectation, we can first condition on the parameters $\Theta_{i-1}$ and the generated context $(X_\tau, \hat{y}_{<t}^{(i-1)})$, which determine the value of the error $e_{t,v}^{(i-1)}$.

$$\mathbb{E}[e_{t,v}^{(i-1)} g_{t,v}^{(i)}] = \mathbb{E}\left[ \mathbb{E}[e_{t,v}^{(i-1)} g_{t,v}^{(i)} \mid \Theta_{i-1}, X_\tau, \hat{y}_{<t}^{(i-1)}] \right] \tag{26}$$

$$= \mathbb{E}\left[ e_{t,v}^{(i-1)} \cdot \mathbb{E}[g_{t,v}^{(i)} \mid \Theta_{i-1}, X_\tau, \hat{y}_{<t}^{(i-1)}] \right] \tag{27}$$

In equation 27, $e_{t,v}^{(i-1)}$ is treated as a constant inside the inner expectation because it is fully determined by the conditioning variables. Let's denote the inner expectation of the contribution term as $\bar{g}_{t,v}^{(i)} \triangleq \mathbb{E}[g_{t,v}^{(i)} \mid \Theta_{i-1}, X_\tau, \hat{y}_{<t}^{(i-1)}]$. The expression simplifies to $\mathbb{E}[e_{t,v}^{(i-1)} \bar{g}_{t,v}^{(i)}]$. We proceed as follows:

$$\mathbb{E}[e_{t,v}^{(i-1)} \bar{g}_{t,v}^{(i)}] = \mathbb{E}[e_{t,v}^{(i-1)}(e_{t,v}^{(i-1)} - (e_{t,v}^{(i-1)} - \bar{g}_{t,v}^{(i)}))]$$

$$= \mathbb{E}[(e_{t,v}^{(i-1)})^2] - \mathbb{E}[e_{t,v}^{(i-1)}(e_{t,v}^{(i-1)} - \bar{g}_{t,v}^{(i)})]$$

Applying the Cauchy-Schwarz inequality to the second term gives:

$$\left| \mathbb{E}[e_{t,v}^{(i-1)}(e_{t,v}^{(i-1)} - \bar{g}_{t,v}^{(i)})] \right| \leq \sqrt{\mathbb{E}[(e_{t,v}^{(i-1)})^2] \cdot \mathbb{E}[(e_{t,v}^{(i-1)} - \bar{g}_{t,v}^{(i)})^2]}$$

From the definitions in Assumption 1, we recognize that $\mathbb{E}[(e_{t,v}^{(i-1)})^2] = \epsilon_{i-1,t,v}^2 + \sigma_{i-1,t,v}^2$, and the bias term is precisely $\epsilon_{g_{t,v}^{(i)}}^2 = \mathbb{E}[(e_{t,v}^{(i-1)} - \bar{g}_{t,v}^{(i)})^2]$. This establishes a rigorous lower bound for our term of interest:

$$\mathbb{E}[e_{t,v}^{(i-1)} g_{t,v}^{(i)}] \geq \mathbb{E}[(e_{t,v}^{(i-1)})^2] - \sqrt{\mathbb{E}[(e_{t,v}^{(i-1)})^2] \cdot \epsilon_{g_{t,v}^{(i)}}^2} \tag{28}$$

$$= \sqrt{\mathbb{E}[(e_{t,v}^{(i-1)})^2]} \left( \sqrt{\mathbb{E}[(e_{t,v}^{(i-1)})^2]} - \epsilon_{g_{t,v}^{(i)}} \right) \tag{29}$$

$$= \sqrt{\epsilon_{i-1,t,v}^2 + \sigma_{i-1,t,v}^2} \left( \sqrt{\epsilon_{i-1,t,v}^2 + \sigma_{i-1,t,v}^2} - \epsilon_{g_{t,v}^{(i)}} \right) \tag{30}$$

Assumption 1 states that $\epsilon_{g_{t,v}^{(i)}} < \sqrt{\epsilon_{i-1,t,v}^2 + \sigma_{i-1,t,v}^2}$, which guarantees that the term in the parentheses is strictly positive. Therefore, $\mathbb{E}[e_{t,v}^{(i-1)} g_{t,v}^{(i)}]$ is strictly positive. As $\mathbb{E}[e_{t,v}^{(i-1)} g_{t,v}^{(i)}] > 0$, the change in MSE, $\Delta_{\mathrm{MSE},t,v}$, will be negative for any $\lambda_i$ in an interval $(0, \lambda_{i,t,v}^*)$ for some positive constant $\lambda_{i,t,v}^*$. This proves the theorem. $\qquad\square$

## I.4  PROOF OF COROLLARY 1

**Corollary 2** (Restate). *For each stage $i \geq 1$ and decoding step $t$, there exists a single, per-token scalar $\lambda_{i,t}^* > 0$, such that for any $\lambda_i \in (0, \lambda_{i,t}^*)$, the total MSE of the new ensemble $\mathcal{M}_i$ is strictly less than that of the predecessor ensemble $\mathcal{M}_{i-1}$.*

*Proof.* From Theorem 1, for each dimension $v \in \{1, \ldots, |\mathcal{V}|\}$, there exists an upper bound $\lambda_{i,t,v}^* > 0$. We define a per-token upper bound $\lambda_{i,t}^*$ as the minimum of these per-dimension bounds:

$$\lambda_{i,t}^* \triangleq \min_{v=1,\ldots,|\mathcal{V}|} \{\lambda_{i,t,v}^*\}. \tag{31}$$

Since this is the minimum over a finite set of strictly positive numbers, $\lambda_{i,t}^*$ is itself strictly positive.

For any scaling coefficient $\lambda_i$ such that $0 < \lambda_i < \lambda_{i,t}^*$, the condition $\lambda_i < \lambda_{i,t,v}^*$ holds for all dimensions $v$. By Theorem 1, this guarantees that the change in MSE for each dimension is negative:

$$\Delta_{\mathrm{MSE},t,v} < 0 \quad \text{for all } v \in \{1, \ldots, |\mathcal{V}|\}. \tag{32}$$

The total change in MSE at step $t$ is the sum of these per-dimension changes:

$$\Delta_{\mathrm{MSE},t} = \mathrm{MSE}(\mathcal{M}_i; t) - \mathrm{MSE}(\mathcal{M}_{i-1}; t) = \sum_{v=1}^{|\mathcal{V}|} \Delta_{\mathrm{MSE},t,v}. \tag{33}$$

As the sum of strictly negative terms, the total change $\Delta_{\mathrm{MSE},t}$ must be strictly negative. Therefore, for any $\lambda_i \in (0, \lambda_{i,t}^*)$, we have $\mathrm{MSE}(\mathcal{M}_i; t) < \mathrm{MSE}(\mathcal{M}_{i-1}; t)$. This concludes the proof. $\qquad\square$

### I.5 Technical Lemmas and Bounds for the Proofs

This appendix provides rigorous justifications for the technical claims made in the proof of Lemma 1.

#### I.5.1 Quadratic Bound on the Taylor Remainder of Softmax

**Definition 1** (Hessian as a Symmetric Bilinear Form). *Let $f : \mathbb{R}^n \to \mathbb{R}^m$ be a twice continuously differentiable function ($C^2$). The second derivative of $f$ at a point $u \in \mathbb{R}^n$, denoted as $D^2 f(u)$, is a bilinear form that maps two vectors from $\mathbb{R}^n$ to a vector in $\mathbb{R}^m$.*

*For any two vectors $v, w \in \mathbb{R}^n$, the resulting vector $D^2 f(u)[v, w] \in \mathbb{R}^m$ is defined component-wise:*

$$\left( D^2 f(u)[v, w] \right)_i = \sum_{j=1}^{n} \sum_{k=1}^{n} \frac{\partial^2 f_i(u)}{\partial u_j \partial u_k} v_j w_k, \quad \text{for } i = 1, \ldots, m.$$

*Since $f$ is $C^2$, by Clairaut's theorem on the equality of mixed partials, this bilinear form is symmetric, i.e., $D^2 f(u)[v, w] = D^2 f(u)[w, v]$. In the context of the Taylor expansion remainder, this form is evaluated with identical vectors, as in $D^2 f(u)[\Delta z, \Delta z]$.*

**Definition 2** (Operator Norm of the Hessian Tensor). *The operator norm of the second derivative $D^2 f(u)$, induced by the vector $\ell_2$-norm, is defined as:*

$$\|D^2 f(u)\|_{\mathrm{op}} = \sup_{\|v\|_2 = 1, \|w\|_2 = 1} \|D^2 f(u)[v, w]\|_2.$$

*For any fixed $u$ in a finite-dimensional space, this norm is finite. The crucial property for the Taylor remainder bound is the **global boundedness** of this norm, i.e., the existence of a constant $L$ such that:*

$$L := \sup_{u \in \mathbb{R}^n} \|D^2 f(u)\|_{\mathrm{op}} < \infty.$$

The global boundedness of the Hessian for $f(z) = \mathrm{softmax}(z)$ is established by observing that its components are polynomials of the outputs $p_i = \mathrm{softmax}(z)_i$. The first and second derivatives of the softmax function are given by:

$$\frac{\partial p_i}{\partial z_j} = p_i(\delta_{ij} - p_j) \tag{34}$$

$$\frac{\partial^2 p_i}{\partial z_j \partial z_k} = p_i \big[ (\delta_{ik} - p_k)(\delta_{ij} - p_j) - p_j(\delta_{jk} - p_k) \big] \tag{35}$$

where $p = \mathrm{softmax}(z)$ and $\delta_{ij}$ is the Kronecker delta.

The components of the Hessian tensor are evidently polynomials in the components of $p$. The image of the softmax function is the open standard simplex. Since these polynomial expressions are continuous, they are bounded on the closure of this image, which is the standard simplex—a compact set. This ensures that each component of the Hessian tensor is globally bounded, which in turn guarantees that its operator norm is globally bounded by a constant $L < \infty$.

**Lemma 2** (Quadratic Bound on the Taylor Remainder of Softmax). *The Taylor remainder $R(\Delta z) = \mathrm{softmax}(z + \Delta z) - \mathrm{softmax}(z) - J_{\mathrm{softmax}}(z)\Delta z$ satisfies the quadratic bound $\|R(\Delta z)\|_2 \leq C\|\Delta z\|_2^2$ for a global constant $C$.*

*Proof.* Let $f(z) = \text{softmax}(z)$. Since $f$ is infinitely differentiable ($C^\infty$), we can express the remainder term using Taylor's theorem with integral remainder:

$$R(\Delta z) = \int_0^1 (1-t) D^2 f(z + t\Delta z)[\Delta z, \Delta z] dt,$$

where $J_f(z)$ is the Jacobian matrix of $f$ at $z$, and $D^2 f(u)[\cdot, \cdot]$ is the second derivative (a bilinear form corresponding to the Hessian tensor) at a point $u$. By taking the vector $\ell_2$-norm, we get:

$$\|R(\Delta z)\|_2 \leq \left( \int_0^1 (1-t) dt \right) \cdot \sup_{t \in [0,1]} \|D^2 f(z+t\Delta z)\|_{\text{op}} \cdot \|\Delta z\|_2^2 = \frac{1}{2} \sup_{u \in [z, z+\Delta z]} \|D^2 f(u)\|_{\text{op}} \cdot \|\Delta z\|_2^2,$$

where $\|\cdot\|_{\text{op}}$ is the operator norm. The entries of the Jacobian and the Hessian of the softmax function are polynomials of its components, $p_i = \text{softmax}(z)_i$. Since the output vector $p$ is always on the standard simplex (i.e., $p_i \in [0,1]$ and $\sum_i p_i = 1$), which is a compact set, the values of these derivatives are globally bounded for any input $z \in \mathbb{R}^{|\mathcal{V}|}$. Therefore, there exists a global constant $L = \sup_{u \in \mathbb{R}^{|\mathcal{V}|}} \|D^2 f(u)\|_{\text{op}} < \infty$. This gives the desired quadratic bound with $C = L/2$, formally justifying that $\|R(\Delta z)\|_2 = O(\|\Delta z\|_2^2)$. $\qquad\square$

### I.5.2    COMPONENT BOUND FROM VECTOR NORM BOUND

**Lemma 3** (Component Bound from Vector Norm Bound). *For any vector $v \in \mathbb{R}^d$, each component $v_m$ is bounded by its $\ell_2$-norm: $|v_m| \leq \|v\|_2$. Consequently, if $\|R(\Delta z)\|_2 = O(\|\Delta z\|_2^2)$, then each component $R_m(\Delta z)$ is also $O(\|\Delta z\|_2^2)$.*

*Proof.* The proof is a direct application of the definition of the $\ell_2$-norm: $\|v\|_2 = \sqrt{\sum_{i=1}^d v_i^2}$. Since all terms in the sum are non-negative, we have $\sqrt{v_m^2} \leq \sqrt{\sum_{i=1}^d v_i^2}$, which simplifies to $|v_m| \leq \|v\|_2$. Applying this to the remainder vector $R$, we get $|R_m| \leq \|R\|_2$. From Lemma 2, there exists a constant $C$ such that $\|R(\Delta z)\|_2 \leq C\|\Delta z\|_2^2$. Therefore, $|R_m(\Delta z)| \leq C\|\Delta z\|_2^2$, which means $R_m = O(\|\Delta z\|_2^2)$. $\qquad\square$

### I.6    EXPECTATION ORDERS OF REMAINDER TERMS

**Lemma 4** (Expectation Orders of Remainder Terms). *Under the finite moment conditions in Assumption 1, for any decoding step $t$ and vocabulary dimension $v$, the expectations of terms involving the Taylor remainder component $R_{t,v} = R_v(\lambda_i z_t^{(i)})$ have the following orders of magnitude with respect to the scaling coefficient $\lambda_i$:*

$$\mathbb{E}[e_{t,v}^{(i-1)} R_{t,v}] = O(\lambda_i^2), \tag{36}$$

$$\mathbb{E}[\lambda_i g_{t,v}^{(i)} R_{t,v}] = O(\lambda_i^3), \tag{37}$$

$$\mathbb{E}[R_{t,v}^2] = O(\lambda_i^4). \tag{38}$$

*Proof.* We analyze each term by applying the Cauchy-Schwarz inequality along with bounds on the Taylor remainder and model outputs. Let the full remainder vector be $R_t = R(\lambda_i z_t^{(i)})$.

**Analysis of** $\mathbb{E}[e_{t,v}^{(i-1)} R_{t,v}]$. By the Cauchy-Schwarz inequality, $|\mathbb{E}[e_{t,v}^{(i-1)} R_{t,v}]| \leq \sqrt{\mathbb{E}[(e_{t,v}^{(i-1)})^2] \mathbb{E}[R_{t,v}^2]}$. The term $\mathbb{E}[(e_{t,v}^{(i-1)})^2]$ is the MSE of the predecessor ensemble for this dimension, which is finite ($O(1)$) by our setup. For the remainder term, we use Lemma 3 to state that $|R_{t,v}| \leq \|R_t\|_2$. Subsequently, Lemma 2 provides that $\|R_t\|_2 \leq C\|\lambda_i z_t^{(i)}\|_2^2 = C\lambda_i^2 \|z_t^{(i)}\|_2^2$ for some global constant $C$. Squaring and taking the expectation, we get $\mathbb{E}[R_{t,v}^2] \leq \mathbb{E}[\|R_t\|_2^2] \leq C^2 \lambda_i^4 \mathbb{E}[\|z_t^{(i)}\|_2^4]$. By Assumption 1, $\mathbb{E}[\|z_t^{(i)}\|_2^4]$ is finite. Thus, $\mathbb{E}[R_{t,v}^2] = O(\lambda_i^4)$. Substituting these orders back into the Cauchy-Schwarz inequality gives $|\mathbb{E}[e_{t,v}^{(i-1)} R_{t,v}]| \leq \sqrt{O(1) \cdot O(\lambda_i^4)} = O(\lambda_i^2)$.

**Analysis of $\mathbb{E}[\lambda_i g_{t,v}^{(i)} R_{t,v}]$.** This expectation can be written as $\lambda_i \mathbb{E}[g_{t,v}^{(i)} R_{t,v}]$. Applying the Cauchy-Schwarz inequality, we have $|\lambda_i \mathbb{E}[g_{t,v}^{(i)} R_{t,v}]| \leq \lambda_i \sqrt{\mathbb{E}[(g_{t,v}^{(i)})^2]\mathbb{E}[R_{t,v}^2]}$. From the analysis above, we know $\mathbb{E}[R_{t,v}^2] = O(\lambda_i^4)$. For the term involving $g_{t,v}^{(i)}$, we note that $g_t^{(i)} = J_{\text{softmax}}(\hat{z}_t^{(i-1)})z_t^{(i)}$. The operator norm of the softmax Jacobian is globally bounded by a constant, say $C_J$. Thus, $\|g_t^{(i)}\|_2 \leq C_J \|z_t^{(i)}\|_2$. This implies $\mathbb{E}[(g_{t,v}^{(i)})^2] \leq \mathbb{E}[\|g_t^{(i)}\|_2^2] \leq C_J^2 \mathbb{E}[\|z_t^{(i)}\|_2^2]$. Since the fourth-order moment of $z_t^{(i)}$ is finite, its second-order moment is also finite, making $\mathbb{E}[(g_{t,v}^{(i)})^2]$ a finite constant ($O(1)$). Combining these results, the overall expression is of order $\lambda_i \sqrt{O(1) \cdot O(\lambda_i^4)} = \lambda_i \cdot O(\lambda_i^2) = O(\lambda_i^3)$.

**Analysis of $\mathbb{E}[R_{t,v}^2]$.** As established in the first paragraph of this proof, the bound $|R_{t,v}| \leq C\lambda_i^2 \|z_t^{(i)}\|_2^2$ leads directly to the expectation bound $\mathbb{E}[R_{t,v}^2] \leq C^2 \lambda_i^4 \mathbb{E}[\|z_t^{(i)}\|_2^4]$. Given the finite fourth-order moment assumption on the logits $z_t^{(i)}$, we confirm that $\mathbb{E}[R_{t,v}^2] = O(\lambda_i^4)$. $\qquad\square$

## J  JUSTIFICATION FOR THE CORE ASSUMPTION: AN OPTIMIZATION PERSPECTIVE

The theoretical analysis in the main body (Section 4) is conducted in the inference setting, establishing performance guarantees for the final trained ensemble. That analysis relies on Assumption 1, which posits that each successor model is an effective error corrector. In this section, we provide a supporting analysis from an optimization perspective during the training phase. We demonstrate that our proposed training objective (Eq. 7) naturally steers the model parameters in a direction that encourages the conditions of Assumption 1 to be met. While this analysis is conducted under a teacher-forcing training setting (using ground-truth prefixes), its conclusions provide a strong heuristic justification for the validity of our core assumption in the auto-regressive inference setting.

### J.1  THEORETICAL STATEMENTS FOR THE TRAINING SETTING

This subsection presents the formal statements of the assumption, theorem, and corollary for the training-time setting, which relies on the Teacher Forcing paradigm. The proofs follow the same structure as those for the inference-time analysis presented in the preceding appendix.

For the analysis at a given decoding step $t$, the context is the ground-truth prefix $(X_\tau, y^*_{<t})$. The ground-truth target is the one-hot vector $y^*_t$. The residual error of the ensemble $\mathcal{M}_{i-1}$ is defined as $e^{(i-1)}_t = y^*_t - \hat{P}^{(i-1)}_t$, where $\hat{P}^{(i-1)}_t$ is the probability distribution of the ensemble.

**Assumption 2.** *For any stage $i \geq 1$, any decoding step $t$, and any vocabulary dimension $v \in \{1, \ldots, |\mathcal{V}|\}$, we assume that the successor model $m_i$, when trained, produces an effective linear contribution $g^{(i)}_t$ that is a valid corrective term of the residual error vector $e^{(i-1)}_t$. Specifically, we assume its systematic bias in fitting this residual is less than the total uncertainty of the residual. Formally, given an input context $(X_\tau, y^*_{<t})$:*

$$\epsilon_{g^{(i)}_{t,v}} < \sqrt{\epsilon^2_{i-1,t,v} + \sigma^2_{i-1,t,v}}, \tag{39}$$

*where the bias ($\epsilon^2$) and variance ($\sigma^2$) terms are defined as:*

$$\epsilon^2_{i-1,t,v} := \mathbb{E}_{(X_\tau,Y_\tau)\sim\mathcal{D}}\left[\left(y^*_{t,v} - \mathbb{E}_{\Theta_{i-1}\sim\mathcal{A}_{i-1}}[\hat{P}^{(i-1)}_{t,v} \mid X_\tau, y^*_{<t}]\right)^2\right] < \infty, \tag{40}$$

$$\sigma^2_{i-1,t,v} := \mathbb{E}_{(X_\tau,Y_\tau)\sim\mathcal{D}}\left[\mathrm{Var}_{\Theta_{i-1}\sim\mathcal{A}_{i-1}}[\hat{P}^{(i-1)}_{t,v} \mid X_\tau, y^*_{<t}]\right] < \infty, \tag{41}$$

$$\epsilon^2_{g^{(i)}_{t,v}} := \mathbb{E}_{\substack{\Theta_{i-1}\sim\mathcal{A}_{i-1} \\ (X_\tau,Y_\tau)\sim\mathcal{D}}}\left[\left(e^{(i-1)}_{t,v} - \mathbb{E}_{\theta_i\sim A_i}[g^{(i)}_{t,v} \mid \Theta_{i-1}, X_\tau, y^*_{<t}]\right)^2\right] < \infty. \tag{42}$$

*Furthermore, we assume that the logits generated by the successor model have a finite fourth-order moment, i.e., $\mathbb{E}[\|z^{(i)}_t(\cdot|X_\tau, y^*_{<t})\|^4_2] < \infty$.*

**Theorem 3.** *Under Assumption 2, for each stage $i \geq 1$, decoding step $t$, and vocabulary dimension $v$, there exists an upper bound $\lambda^*_{i,t,v} > 0$. For any scaling coefficient $\lambda_i \in (0, \lambda^*_{i,t,v})$, the expected squared error (MSE) of the new ensemble $\mathcal{M}_i$ is strictly lower than that of the predecessor ensemble $\mathcal{M}_{i-1}$ for that dimension. Specifically:*

$$\mathbb{E}_{\substack{\Theta_{i-1}\sim\mathcal{A}_{i-1} \\ \theta_i\sim A_i \\ (X_\tau,Y_\tau)\sim\mathcal{D}}}\left[\left(y^*_{t,v} - \hat{P}^{(i)}_{t,v}\right)^2 \mid X_\tau, y^*_{<t}\right] < \mathbb{E}_{\substack{\Theta_{i-1}\sim\mathcal{A}_{i-1} \\ (X_\tau,Y_\tau)\sim\mathcal{D}}}\left[\left(y^*_{t,v} - \hat{P}^{(i-1)}_{t,v}\right)^2 \mid X_\tau, y^*_{<t}\right]. \tag{43}$$

**Corollary 3.** *For each stage $i \geq 1$ and decoding step $t$, there exists a single, per-token scalar $\lambda^*_{i,t} > 0$, independent of vocabulary dimension. For any scaling coefficient $\lambda_i \in (0, \lambda^*_{i,t})$, the total MSE of the new ensemble $\mathcal{M}_i$ for that token prediction is strictly less than that of the predecessor ensemble $\mathcal{M}_{i-1}$:*

$$\mathbb{E}_{\substack{\Theta_{i-1}\sim\mathcal{A}_{i-1} \\ \theta_i\sim A_i \\ (X_\tau,Y_\tau)\sim\mathcal{D}}}\left[\|y^*_t - \hat{P}^{(i)}_t\|^2_2 \mid X_\tau, y^*_{<t}\right] < \mathbb{E}_{\substack{\Theta_{i-1}\sim\mathcal{A}_{i-1} \\ (X_\tau,Y_\tau)\sim\mathcal{D}}}\left[\|y^*_t - \hat{P}^{(i-1)}_t\|^2_2 \mid X_\tau, y^*_{<t}\right]. \tag{44}$$

## J.2 SETUP AND PRELIMINARY LEMMAS

We focus on the training of a single successor model $m_i$ with parameters $\theta_i \in \mathbb{R}^d$ at an arbitrary decoding step $t$.

- **Losses:** The total loss for token $t$ is $\mathcal{L}_{i,t}(\theta_i) = \mathcal{L}_{s,t}(\theta_i) + \alpha\mathcal{L}_{ce,t}(\theta_i)$, where $\mathcal{L}_{s,t}$ is the error-suppression loss and $\mathcal{L}_{ce,t}(\theta_i) = -\log p_t^{(i)}(y_t^*; \theta_i)$ is the cross-entropy loss.
- **Gradients:** Let $g_{s,t}(\theta_i) = \nabla_{\theta_i}\mathcal{L}_{s,t}(\theta_i)$, $g_{ce,t}(\theta_i) = \nabla_{\theta_i}\mathcal{L}_{ce,t}(\theta_i)$, and the total gradient is $g_{total,t}(\theta_i) = g_{s,t}(\theta_i) + \alpha g_{ce,t}(\theta_i)$.
- **Logits and Jacobian:** Let $z_t^{(i)}$ be the logits from model $m_i$. Let $J_i(\theta_i) = \frac{\partial z_t^{(i)}(\theta_i)}{\partial \theta_i}$ be the Jacobian of the logit function.

We begin by characterizing the exact gradients of the loss components.

**Lemma 5** (Loss Gradients in Logit Space). *At a given step t, let $p_t^{(i)} = \mathrm{softmax}(z_t^{(i)})$. Let $y_t^*$ be the one-hot vector for the correct token, and if the predecessor erred, let $y_{err}$ be the one-hot vector for the $\mathrm{ErrorToken}_t^{(i-1)}$. The gradients of the loss components with respect to the logits $z_t^{(i)}$ are:*

$$\nabla_{z_t^{(i)}}\mathcal{L}_{ce,t} = p_t^{(i)} - y_t^*, \tag{45}$$

$$\nabla_{z_t^{(i)}}\mathcal{L}_{s,t} = \beta\sigma(-u(\theta_i))(y_{err} - y_t^*) \cdot \mathbf{1}\{\mathrm{ErrorToken}_t^{(i-1)} \neq \emptyset\}, \tag{46}$$

*where $u(\theta_i) = \beta[\log p_t^{(i)}(y_t^*) - \log p_t^{(i)}(\mathrm{ErrorToken}_t^{(i-1)})]$.*

*Proof.* The gradient for $\mathcal{L}_{ce,t}$ is a standard result. For $\mathcal{L}_{s,t}$, we apply the chain rule $\nabla_z\mathcal{L}_{s,t} = \frac{\partial \mathcal{L}_{s,t}}{\partial u}\frac{\partial u}{\partial z}$. We have $\frac{\partial \mathcal{L}_{s,t}}{\partial u} = -\sigma(-u)$. For the second term:

$$\frac{\partial u}{\partial z_t^{(i)}} = \beta\left(\frac{\partial \log p_t^{(i)}(y_t^*)}{\partial z_t^{(i)}} - \frac{\partial \log p_t^{(i)}(y_{err})}{\partial z_t^{(i)}}\right). \tag{47}$$

Using the identity $\frac{\partial \log p_k}{\partial z} = \mathbf{e}_k - p$, where $\mathbf{e}_k$ is the one-hot vector for class $k$:

$$\frac{\partial u}{\partial z_t^{(i)}} = \beta\left((y_t^* - p^{(i)}) - (y_{err} - p^{(i)})\right) = \beta(y_t^* - y_{err}). \tag{48}$$

Combining these gives $\nabla_{z_t^{(i)}}\mathcal{L}_{s,t} = -\sigma(-u) \cdot \beta(y_t^* - y_{err}) = \beta\sigma(-u)(y_{err} - y_t^*)$, which proves the result for the case when the predecessor errs. Otherwise, the gradient is zero. $\square$

**Corollary 4** (Loss Gradients in Parameter Space). *The gradients with respect to the parameters $\theta_i$ are obtained via the Jacobian: $g_{ce,t}(\theta_i) = J_i(\theta_i)^\top(p_t^{(i)} - y_t^*)$ and $g_{s,t}(\theta_i) = J_i(\theta_i)^\top\left[\beta\sigma(-u)(y_{err} - y_t^*) \cdot \mathbf{1}\{\mathrm{ErrorToken}_t^{(i-1)} \neq \emptyset\}\right].$*

## J.3 GRADIENT ALIGNMENT AND MAIN RESULT

To formalize the argument that our training objective provides a valid descent path, we introduce two standard assumptions regarding the optimization landscape. We analyze the gradients of the primary cross-entropy loss, $g_{ce,t}(\theta_i)$, and the error-suppression loss, $g_{s,t}(\theta_i)$. The total gradient used for the update is $g_{total,t}(\theta_i) = \alpha g_{ce,t}(\theta_i) + g_{s,t}(\theta_i)$.

**Assumption 3** (Smoothness and Parameter Compactness). *The primary loss function, $\mathcal{L}_{ce,t}(\theta_i)$, is L-smooth for some constant $L > 0$, meaning its gradient is Lipschitz continuous with constant $L$. Furthermore, we assume the training trajectory of the parameters $\theta_i$ remains within a compact set.*

**Assumption 4** (Gradient Relative Alignment and Bounded Norm). *During the training of successor $m_i$, for the subset of training data where the predecessor model $m_{i-1}$ predicts incorrectly, we assume there exist constants $\rho_i \in [0, 1)$ and $\Gamma_i > 0$ that characterize the geometric relationship between the two gradient components for all parameters $\theta_i$ encountered during training:*

$$\langle g_{s,t}(\theta_i), g_{ce,t}(\theta_i)\rangle \geq -\rho_i\|g_{s,t}(\theta_i)\|\|g_{ce,t}(\theta_i)\|, \tag{49}$$

$$\|g_{s,t}(\theta_i)\| \leq \Gamma_i\|g_{ce,t}(\theta_i)\|. \tag{50}$$

**Remark.** The first condition, relative alignment, bounds the degree to which the error-suppression loss can oppose the primary training signal. A value of $\rho_i$ close to 0 indicates strong alignment. The second condition bounds the relative magnitude of the two gradients. Together, they formalize the notion that $\mathcal{L}_{s,t}$ acts as a helpful, non-contradictory regularizer.

**Theorem 4** (Guaranteed Descent on Primary Loss). *Under Assumptions 3 and 4, there exists a constructive range for the hyperparameter $\alpha$ and the learning rate $\eta$ that guarantees a decrease in the primary cross-entropy loss $\mathcal{L}_{ce,t}$ with each gradient descent step $\theta_i' = \theta_i - \eta g_{total,t}(\theta_i)$, provided $g_{ce,t}(\theta_i) \neq 0$.*

*Proof.* The $L$-smoothness of $\mathcal{L}_{ce,t}$ provides the following descent lemma:

$$\mathcal{L}_{ce,t}(\theta_i') \leq \mathcal{L}_{ce,t}(\theta_i) - \eta\langle g_{total,t}(\theta_i), g_{ce,t}(\theta_i)\rangle + \frac{L\eta^2}{2}\|g_{total,t}(\theta_i)\|^2. \tag{51}$$

To guarantee a decrease in $\mathcal{L}_{ce,t}$, a sufficiently small learning rate $\eta$ can be chosen provided that the inner product $\langle g_{total,t}, g_{ce,t}\rangle$ is strictly positive. We analyze this inner product based on the predecessor's prediction.

**Case 1: Predecessor is correct** ($\text{ErrorToken}_t^{(i-1)} = \emptyset$). In this case, $g_{s,t}(\theta_i) = 0$. The total gradient simplifies to $g_{total,t}(\theta_i) = \alpha g_{ce,t}(\theta_i)$. The inner product becomes:

$$\langle g_{total,t}, g_{ce,t}\rangle = \langle \alpha g_{ce,t}, g_{ce,t}\rangle = \alpha\|g_{ce,t}\|^2. \tag{52}$$

Since we assume $g_{ce,t} \neq 0$ and $\alpha > 0$, this inner product is strictly positive.

**Case 2: Predecessor is incorrect** ($\text{ErrorToken}_t^{(i-1)} \neq \emptyset$). The full gradient is active. We must find conditions under which the full inner product is positive.

**Step 1: Lower bound the inner product** $\langle g_{total,t}, g_{ce,t}\rangle$.

$$\langle g_{total,t}, g_{ce,t}\rangle = \langle \alpha g_{ce,t} + g_{s,t}, g_{ce,t}\rangle = \alpha\|g_{ce,t}\|^2 + \langle g_{s,t}, g_{ce,t}\rangle \tag{53}$$

$$\geq \alpha\|g_{ce,t}\|^2 - \rho_i\|g_{s,t}\|\|g_{ce,t}\| \quad \text{(by Eq. 49)} \tag{54}$$

$$\geq \alpha\|g_{ce,t}\|^2 - \rho_i(\Gamma_i\|g_{ce,t}\|)\|g_{ce,t}\| \quad \text{(by Eq. 50)} \tag{55}$$

$$= (\alpha - \rho_i\Gamma_i)\|g_{ce,t}\|^2. \tag{56}$$

For this inner product to be strictly positive, we require $\alpha - \rho_i\Gamma_i > 0$. This gives us a lower bound for our hyperparameter $\alpha$:

$$\alpha > \rho_i\Gamma_i.$$

**Step 2: Upper bound the total gradient norm** $\|g_{total,t}\|^2$.

$$\|g_{total,t}\|^2 = \|\alpha g_{ce,t} + g_{s,t}\|^2 \leq (\|\alpha g_{ce,t}\| + \|g_{s,t}\|)^2 \tag{57}$$

$$\leq (\alpha\|g_{ce,t}\| + \Gamma_i\|g_{ce,t}\|)^2 \quad \text{(by Eq. 50)} \tag{58}$$

$$= (\alpha + \Gamma_i)^2\|g_{ce,t}\|^2. \tag{59}$$

**Step 3: Construct the learning rate bound.** With a choice of $\alpha > \rho_i\Gamma_i$, the inner product $\langle g_{total,t}, g_{ce,t}\rangle$ is guaranteed to be positive. To ensure descent, the learning rate $\eta$ must be chosen in the interval:

$$0 < \eta < \frac{2\langle g_{total,t}, g_{ce,t}\rangle}{L\|g_{total,t}\|^2}. \tag{60}$$

To find a constructive, safe range for $\eta$, we can establish a lower bound for the right-hand side of the inequality. Using the lower bound for the numerator (from Step 1) and the upper bound for the denominator's norm term (from Step 2), we have:

$$\frac{2\langle g_{total,t}, g_{ce,t}\rangle}{L\|g_{total,t}\|^2} \geq \frac{2(\alpha - \rho_i\Gamma_i)\|g_{ce,t}\|^2}{L(\alpha + \Gamma_i)^2\|g_{ce,t}\|^2} = \frac{2(\alpha - \rho_i\Gamma_i)}{L(\alpha + \Gamma_i)^2}. \tag{61}$$

Let us define this constructive lower bound as $\eta^*(\alpha) = \frac{2(\alpha - \rho_i\Gamma_i)}{L(\alpha + \Gamma_i)^2}$. For any $\alpha > \rho_i\Gamma_i$, the numerator is positive, so $\eta^*(\alpha) > 0$. Therefore, by choosing any learning rate $\eta$ such that $0 < \eta \leq \eta^*(\alpha)$, we guarantee that it is within the valid range for descent, ensuring that $\mathcal{L}_{ce,t}(\theta_i') < \mathcal{L}_{ce,t}(\theta_i)$.

By combining both cases, we have shown that a valid range for $\alpha$ and $\eta$ exists to ensure descent on the primary cross-entropy loss. $\square$

## J.4 CONNECTING THE OPTIMIZATION VIEW TO THE CORE ASSUMPTION

The results from this optimization analysis provide a strong heuristic basis for Assumption 2. Theorem 4 demonstrates that our composite loss function, under the geometric conditions of Assumption 4, is designed to effectively minimize the primary cross-entropy loss $\mathcal{L}_{ce,t}$. A reduction in $\mathcal{L}_{ce,t}$ implies that the model's prediction $p_t^{(i)}$ moves closer to the ground truth $y_t^*$. This is the fundamental mechanism that drives the model $m_i$ to become an effective corrector of the predecessor's residual error, thereby satisfying the conditions required for the MSE reduction theorems.

