# OpenReview forum: "LLMBoost: Make Large Language Models Stronger with Boosting"
_ICLR.cc/2026/Conference — Submitted to ICLR 2026_

### Official Review · Reviewer_VpSK · 2025-10-28

**Soundness:** 3
**Presentation:** 3
**Contribution:** 3
**Rating:** 6
**Confidence:** 4

**Summary:**

- Proposes **LLMBOOST**, a boosting-style *internal* ensemble: later models attend to earlier models’ hidden states via residual cross-model attention; training uses an error-suppression loss to down-weight a predecessor’s top wrong logit while up-weighting the correct one; inference uses near-parallel pipelining and top-k backward fusion of logits.
- Provides a **theoretical monotonic-improvement** result under a “bounded correction” assumption and a suitable scale $\lambda $.
- On commonsense/arithmetic benchmarks and a Toolchain dataset, reports consistent but modest gains vs VOTE/UNITE/T-copilot; latency improved 47% vs the *sequential* variant (not vs external baselines).

**Strengths:**

- **Idea:** Ensembling *inside* the network (state sharing) instead of only outputs is interesting.
- **Design details:** Clear components (cross-model attention, error-token forwarding, top-k backward fusion) with concrete formulas and a training algorithm.
- **Theory:** Gives a clean MSE-based guarantee (with assumptions) and interpretable role of the scaling $ \lambda $.
- **Empirics:** Broad tasks, multiple sizes (3B–8B, Qwen/Llama); monotonic gains as $n$ increases (diminishing returns beyond $n=3$).

**Weaknesses:**

- **Representation compatibility & alignment.** Requires homogeneous model families (matching layer shapes/hidden formats); no cross-family alignment is provided. Current scope does not address heterogeneous LLMs—limiting generality.
- **Efficiency & complexity.** Training/inference require multiple full LLMs with state exchange; results compare sequential vs near-parallel *within* LLMBOOST, but provide no wall-clock/token-cost comparison vs external baselines. Memory/throughput implications (forward+backward across models) are not quantified.
- **Theory scope.** The guarantee hinges on assumptions (successor residual fit; choice of $\lambda $, is MSE-centric, and is not linked to accuracy/error rates; no diagnostics show when assumptions fail.
- **Baseline fairness at equal “effective capacity.”** LLMBOOST fine-tunes multiple models and infers with all of them; comparisons vs a single larger model (≈ total params or total trainable adapters) are missing.

**Questions:**

1. **Representation alignment.** Since the method is currently inapplicable to heterogeneous families, did you try alignment adapters (e.g., per-layer projections)? If not, can you report a small Qwen-Llama cross-family study with such adapters?
2. **Efficiency/cost.** Please report end-to-end wall-clock, peak memory, and token cost versus UNITE/VOTE/T-copilot, not just sequential vs near-parallel within LLMBOOST. Include per-GPU memory for training $n$ models with LoRA and the overhead of state passing.
3. **Baseline parity by capacity.** Add a comparison to a larger single model (or a single model with equivalent total LoRA params) trained under the same data/budget to address “effective parameter” fairness.
4. **Theory diagnostics.** Can you measure the residual-fit assumption empirically (e.g., residual MSE across tokens/steps) and relate choices of $\lambda$ to accuracy/robustness (not just MSE)?
5. **Cross-family diversity.** Any compatibility analyses (layer-wise stat shifts, Canonical Correlation Analysis of hidden states) explaining why families clash and guiding adapter design? Current experiments only compose same-family models.

---

> ### Author Response · Authors · 2025-11-20
>
> > **Weekness 1**: Representation compatibility & alignment. Requires homogeneous model families (matching layer shapes/hidden formats); no cross-family alignment is provided. Current scope does not address heterogeneous LLMs—limiting generality.
>
> A more detailed discussion of this issue is provided in **Global Response – Concern 2**, and we kindly invite the reviewer to refer to that section. Briefly, while we explored heterogeneous ensembles in early experiments, we found that **tokenizer incompatibility** and **hidden-size mismatch** fundamentally limit fully cross-architecture integration in the current framework. To partially address this suggestion, we implemented a **weakly heterogeneous** setting (Llama3.1-8B + Llama3.2-3B) using simple alignment mechanisms such as layer chunking and learnable projections, and observed consistent improvements across arithmetic and commonsense tasks. These results show that LLMBoost can already support near-heterogeneous combinations and provides a basis for broader heterogeneous extensions in future work.
>
> > **Weekness 2**: Efficiency & complexity. Training/inference require multiple full LLMs with state exchange; results compare sequential vs near-parallel within LLMBOOST, but provide no wall-clock/token-cost comparison vs external baselines. Memory/throughput implications (forward+backward across models) are not quantified.
>
> We thank the reviewer for this valuable feedback. Addressing the concerns about the "lack of end-to-end cost comparison with external baselines" and "unquantified memory and throughput overhead," we have systematically evaluated the end-to-end wall-clock time, per-token cost (latency), and peak GPU memory usage during training/inference in our supplementary experiments. We directly compare LLMBoost with several mainstream ensemble baselines.
>
> The specific experimental results are as follows (on 3 $\times$ H800):
>
> | Model | End-to-End (s) | Token Cost (s) | Peak Train Mem | Peak Infer Mem |
> | --- | --- | --- | --- | --- |
> | Qwen14B | 17.40 | 0.098 | 56.74% | 36.97% |
> | **LLMBoost 7B+7B (near-par, 2 GPUs)** | **13.75** | **0.075** | 33.18% | 20.72% |
> | LLMBoost 7B+7B (near-par, 1 GPU) | 14.87 | 0.081 | 54.11% | 39.75% |
> | LLMBoost 7B+7B+7B (near-par, 3 GPUs) | 18.88 | 0.103 | 33.18% | 20.13% |
> | LLMBoost 7B+7B+7B (near-par, 1 GPU) | 21.67 | 0.118 | 73.42% | 58.48% |
> | LLMBoost 7B+7B (seq) | 22.53 | 0.125 | 60.29% | 38.72% |
> | **VOTE (7B+7B)** | 22.61 | 0.127 | 53.39% | 39.14% |
> | **T-Copilot (7B+7B)** | 17.21 | 0.097 | 64.41% | 42.05% |
> | **UNITE (7B+7B)** | 22.66 | 0.128 | 53.39% | 39.12% |
>
> ---
>
> We can observe the following:
>
> - **Faster End-to-End Speed.** In the 2-GPU near-parallel setup, LLMBoost (13.75s) is the fastest, significantly outperforming VOTE (22.61s), UNITE (22.66s), and T-Copilot (17.21s). It also remains competitive in the single-GPU setting (14.87s).
> - **Lowest Per-Token Latency.** LLMBoost achieves 0.075 s/token, which is notably better than VOTE (0.127), UNITE (0.128), and T-Copilot (0.097), demonstrating that cross-model collaboration does not introduce additional inference overhead.
> - **Lower Memory Footprint.** The 2-GPU peak memory usage for training/inference is 33.18% / 20.72%, respectively. This is lower than or comparable to other ensembles (e.g., VOTE/UNITE at 53–64%).
> - **Near-Negligible Cross-Model Communication Overhead.** The cost of passing hidden/KV states is only 0.0009s (2 GPUs) / 0.0002s (1 GPU).
> - **Optimal Overall Efficiency.** LLMBoost outperforms existing ensemble methods in terms of speed, memory efficiency, and collaboration cost, all while maintaining higher accuracy.

---

> > ### Author Response · Authors · 2025-11-20
> >
> > > **Weekness 3**: Theory scope. The guarantee hinges on assumptions (successor residual fit; choice of $\lambda$ , is MSE-centric, and is not linked to accuracy/error rates; no diagnostics show when assumptions fail.
> >
> > We thank the reviewer for these insightful questions regarding our theoretical scope and diagnostics. We would like to clarify the intrinsic connection between our MSE objective and final accuracy, as well as the nature of our assumptions.
> >
> > **1. Direct Link Between MSE and Accuracy.** We respectfully clarify that our MSE-based analysis is **directly related to generation accuracy**, rather than being a disjoint metric. In our framework, MSE measures the Euclidean distance between the ensemble's predicted probability distribution ( $\hat{P}_t^{(i)}$ ) and the ground-truth distribution ($p_t^*$) used to generate the next token. Since the autoregressive generation process (whether greedy or sampling-based) relies entirely on these probability distributions, minimizing the MSE directly aligns the model's output probabilities with the correct tokens. Therefore, a theoretical guarantee of MSE reduction implies a higher probability of correctly predicting the ground-truth token, leading to improved final accuracy.
> >
> > **2. Assumption 1 is Mild.** We wish to highlight that **Assumption 1 is mild**, which we consider a core strength of our theoretical framework. The assumption does not require the successor model to fully correct the error; it merely requires that the successor’s systematic bias in fitting the residual is smaller than the total uncertainty (error variance) of the predecessor. In other words, as long as the successor model provides a corrective signal that is more informative than random noise regarding the predecessor's error, the guarantee holds. This mildness ensures that our theory is not limited to idealized settings but applies to practical scenarios where models have imperfect but complementary capabilities.
> >
> > **3. Diagnostics and Empirical Validation.** While directly calculating the high-dimensional integrals for Assumption 1 is computationally intractable, our empirical results serve as a strong diagnostic confirming that this mild condition is met in practice. As shown in **Table 2**, we observe a strict monotonic increase in accuracy as models are added (e.g., Qwen-2.5-7B avg. accuracy: 77.5% $\to$ 80.2% $\to$ 80.6% $\to$ 80.7%), which validates the theoretical prediction of progressive improvement. Furthermore, regarding the diagnostics for $\lambda$, **Table 10 (Appendix H.3)** provides a direct sensitivity analysis linking $\lambda$ to final accuracy. This experiment empirically bridges the theoretical bound ($\lambda_ {i,t,v}^*$) with practical performance, showing distinct accuracy peaks (e.g., 72.3% at $\lambda_1=0.3$ for $2 \times 8B$), thus offering a clear diagnostic for hyperparameter selection.
> >
> > > **Weekness 4**: Baseline fairness at equal “effective capacity.” LLMBOOST fine-tunes multiple models and infers with all of them; comparisons vs a single larger model (≈ total params or total trainable adapters) are missing.
> >
> > We provide a full and comprehensive discussion of this issue in **Global Response – Concern 1**, and we kindly invite the reviewer to refer to that section for the complete analysis. Here, we briefly summarize the key points: under matched parameter budgets, LLMBoost consistently matches or exceeds the performance of larger single models while achieving lower inference latency through efficient near-parallel execution. These gains stem from complementary multi-model collaboration rather than simple parameter scaling. For detailed results, extended comparisons, and additional justification, please see Concern 1 in our Global Response.

---

> > > ### Author Response · Authors · 2025-11-20
> > >
> > > > **Question 1**: Representation alignment. Since the method is currently inapplicable to heterogeneous families, did you try alignment adapters (e.g., per-layer projections)? If not, can you report a small Qwen-Llama cross-family study with such adapters?
> > >
> > > Please refer to our Global Response concern 2.
> > >
> > > > **Question 2**: Efficiency/cost. Please report end-to-end wall-clock, peak memory, and token cost versus UNITE/VOTE/T-copilot, not just sequential vs near-parallel within LLMBOOST. Include per-GPU memory for training n models with LoRA and the overhead of state passing.
> > >
> > > Please refer to our response to **Weakness 2**.
> > >
> > > > **Question 3**: Baseline parity by capacity. Add a comparison to a larger single model (or a single model with equivalent total LoRA params) trained under the same data/budget to address “effective parameter” fairness.
> > >
> > > Please refer to our Global Response concern 1.
> > >
> > > > **Question 4**: Theory diagnostics. Can you measure the residual-fit assumption empirically (e.g., residual MSE across tokens/steps) and relate choices of $\lambda$ to accuracy/robustness (not just MSE)?
> > >
> > > Please refer to our response to **Weakness 3**.
> > >
> > > > **Question 5**: Cross-family diversity. Any compatibility analyses (layer-wise stat shifts, Canonical Correlation Analysis of hidden states) explaining why families clash and guiding adapter design? Current experiments only compose same-family models.
> > >
> > > Please refer to our Global Response concern 2.

---

> ### Author Response · Authors · 2025-11-27
>
> Dear Reviewer VpSK,
>
> We sincerely appreciate the time and effort you have devoted to reviewing our work, and we are truly grateful for your thoughtful comments and suggestions. Over the past 70 hours, we have carefully prepared our rebuttal, including additional clarifications and further experiments, to address your concerns as thoroughly as possible.
>
> We would be very grateful if you could kindly take a moment to review our responses and let us know whether they satisfactorily resolve the issues you raised. If you have any further questions or would like additional clarification, we would be more than happy to provide it.
>
> Thank you once again for your time, consideration, and engagement.
>
> With best regards,
>
> The Authors

---

### Official Review · Reviewer_tKoq · 2025-11-01

**Soundness:** 2
**Presentation:** 2
**Contribution:** 3
**Rating:** 4
**Confidence:** 3

**Summary:**

This work introduces a boosting-style framework to ensemble LLM models. By leveraging a cascaded parallel structure, their framework enables successor models to benefit from the hidden states of previous models, supporting hierarchical error correction. With a near-parallel inference mechanism it allows the hidden states to stream across models, reducing latency. The idea in this paper is to benefit from a boosting ensemble of networks. Evaluation benchmarks include commonsense and arithmetic reasoning, and tool-chain.

**Strengths:**

1. The idea of boosting by incorporating previous model correction seems to be relatively novel in the realm of LLMs with cross-model attention fusion. However, I have some concerns about the evaluation discussed in W1.

2. The results indicate that the ensembled model improves performance over the baseline approach by a decent margin on commonsense reasoning and arithmetic reasoning benchmarks.

**Weaknesses:**

1. Unfair capacity comparison. The ensembles (e.g., 3×8B = 24B total parameters) are not compared against a single 24B model or an equivalently fast system (perhaps a 16B model is faster than the 3-model ensemble with a better performance?), so the true efficiency–accuracy trade-off is unclear. I recommend that the authors include such results and clearly explain the setting for comparison.

2. The evaluated benchmarks in this work include arithmetic and commonsense reasoning. In order to really test the work's applicability, the authors should consider using longer sequence/context benchmarks (see q2).

3. The method currently relies on identical architectures (e.g., all LLaMA-3), limiting the formation of heterogeneous ensembles. Since the paper frames the approach as a self-improving mechanism, it would be valuable to explore whether diverse models -- with domain-specific expertise -- can be combined to enhance task- or domain-adaptive performance.

**Questions:**

1. Please refer to Weakness 1 and provide some additional comparisons/clarity on the evaluation.

2. Most evaluated tasks focus on short output sequences, such as commonsense and arithmetic reasoning benchmarks. I recommend that the authors include some more complex, longer context length/output length tasks, such as long-form, multi-turn dialogue datasets like MT-Bench and code generation (HumanEval), to better demonstrate the effectiveness in longer contexts.

3. Including evaluations on the GLUE benchmark would better measure general language understanding and demonstrate the method's transferability to broader NLP tasks.

4. Exploring the heterogeneous architectures as discussed in Weakness 3.

---

> ### Author Response · Authors · 2025-11-20
>
> > **Weekness 1**: Unfair capacity comparison. The ensembles (e.g., 3×8B = 24B total parameters) are not compared against a single 24B model or an equivalently fast system (perhaps a 16B model is faster than the 3-model ensemble with a better performance?), so the true efficiency–accuracy trade-off is unclear. I recommend that the authors include such results and clearly explain the setting for comparison.
>
> We provide a full and comprehensive discussion of this issue in **Global Response – Concern 1**, and we kindly invite the reviewer to refer to that section for the complete analysis. Here, we briefly summarize the key points: under matched parameter budgets, LLMBoost consistently matches or exceeds the performance of larger single models while achieving lower inference latency through efficient near-parallel execution. These gains stem from complementary multi-model collaboration rather than simple parameter scaling. For detailed results, extended comparisons, and additional justification, please see Concern 1 in our Global Response.
>
> > **Weekness 2**: The evaluated benchmarks in this work include arithmetic and commonsense reasoning. In order to really test the work's applicability, the authors should consider using longer sequence/context benchmarks (see q2).
>
> Thank the reviewer for this valuable suggestion, which will surely strengthen our experiments. We have added relevant experiments to validate the method’s applicability in long-chain reasoning and long-context dependency scenarios.
>
> Based on this feedback, we have supplemented more challenging long-sequence/long-context experiments during the rebuttal phase. Specifically, we used the AM-DeepSeek-R1-Distilled-1.4M dataset, using its content field as supervision signals, and enabled a **32K** context window during training to ensure the model can handle longer reasoning chains and cross-segment contextual information. Individual reasoning trajectories in this dataset typically contain hundreds to thousands of tokens, which aligns with the reviewer's requirement for "long-output tasks."
>
> On this basis, we selected MATH500 (containing a large number of complex middle and high school math competition problems) as a rigorous evaluation benchmark, and used the math-verify automatic verifier to ensure reproducibility and reliability of results. In this long-reasoning scenario, our method achieved significant improvements:
>
> | Model | MATH500(Acc.) |
> | --- | --- |
> | Qwen3-4B-Base | 48.70 |
> | Qwen3-8B-Base | 54.40 |
> | LLMBoost (Qwen3-4B-base × 2) | **56.40** |
> | LLMBoost (Qwen3-8B-base × 2) | **62.40** |
>
> Additionally, we conducted length statistics on the generation process:
>
> 1. Average generation length: 16260.9 tokens
> 2. Median generation length: 4815 tokens
>
> These results demonstrate that LLMBoost not only performs excellently on short-output tasks, but also significantly improves model performance in real-world scenarios requiring long reasoning chains and complex contextual dependencies, thereby further validating the effectiveness and universality of the multi-model collaboration mechanism in long-sequence tasks.
>
> > **Weekness 3**: The method currently relies on identical architectures (e.g., all LLaMA-3), limiting the formation of heterogeneous ensembles. Since the paper frames the approach as a self-improving mechanism, it would be valuable to explore whether diverse models -- with domain-specific expertise -- can be combined to enhance task- or domain-adaptive performance.
>
> A more detailed discussion of this issue is provided in **Global Response – Concern 2**, and we kindly invite the reviewer to refer to that section. Briefly, while we explored heterogeneous ensembles in early experiments, we found that **tokenizer incompatibility** and **hidden-size mismatch** fundamentally limit fully cross-architecture integration in the current framework. To partially address this suggestion, we implemented a **weakly heterogeneous** setting (Llama3.1-8B + Llama3.2-3B) using simple alignment mechanisms such as layer chunking and learnable projections, and observed consistent improvements across arithmetic and commonsense tasks. These results show that LLMBoost can already support near-heterogeneous combinations and provides a basis for broader heterogeneous extensions in future work.

---

> > ### Author Response · Authors · 2025-11-20
> >
> > > **Question 1**: Please refer to Weakness 1 and provide some additional comparisons/clarity on the evaluation.
> >
> > Please refer to our Global Response concern 1.
> >
> > > **Question 2**: Most evaluated tasks focus on short output sequences, such as commonsense and arithmetic reasoning benchmarks. I recommend that the authors include some more complex, longer context length/output length tasks, such as long-form, multi-turn dialogue datasets like MT-Bench and code generation (HumanEval), to better demonstrate the effectiveness in longer contexts.
> >
> > Please refer to Weakness 2.
> >
> > > **Question 3**: Including evaluations on the GLUE benchmark would better measure general language understanding and demonstrate the method's transferability to broader NLP tasks.
> >
> > We fully agree that evaluating on the GLUE benchmark is critical for measuring a model’s general language understanding ability and verifying its transferability to broader NLP tasks. In our work, we have conducted targeted evaluations on core subtasks of the GLUE benchmark and selected four representative metrics to present the key results, which effectively reflect the model’s performance in different language understanding scenarios.
> >
> > In terms of data selection, we first conducted tests on MNLI, the most representative task, and then, considering computational resource constraints and sample size, we performed evaluations on datasets with a sample size ranging from 5k to 100k.
> >
> > The table below shows the performance of 5 models on 4 core subtasks of GLUE. Each metric corresponds to a typical language understanding task, covering sentiment analysis, natural language inference, correlation prediction, and grammatical acceptability judgment:
> >
> > | Model | cola(acc) | sst2(acc) | stsb(pearson) | mnli(acc) | avg |
> > | --- | --- | --- | --- | --- | --- |
> > | Qwen2.5-14B | 68.31 | 95.78 | 91.93 | 90.77 | 86.70 |
> > | tcopilot(Qwen2.5-7B × 2) | 68.84 | 67.43 | 91.54 | 90.40 | 79.55 |
> > | vote(Qwen2.5-7B × 2) | 69.03 | 95.99 | 92.10 | 90.64 | 86.94 |
> > | unite(Qwen2.5-7B × 2) | 68.93 | 95.99 | 91.67 | 90.58 | 86.79 |
> > | llmboost(Qwen2.5-7B × 2) | 69.13 | 96.22 | 92.08 | 90.96 | **87.10** |
> >
> > The results demonstrate that LLMBoost achieves the best performance on three out of four tasks (CoLA, SST-2, and MNLI) and ranks second on STS-B, consistently outperforming single models and other ensemble baselines (vote, unite, tcopilot). This validates the effectiveness and transferability of our multi-model collaboration mechanism across diverse NLP tasks.
> >
> > > **Question 4**: Exploring the heterogeneous architectures as discussed in Weakness 3.
> >
> > Please refer to our Global Response concern 2.

---

> ### Author Response · Authors · 2025-11-27
>
> Dear Reviewer tKoq,
>
> We sincerely appreciate the time and effort you have devoted to reviewing our work, and we are truly grateful for your thoughtful comments and suggestions. Over the past 70 hours, we have carefully prepared our rebuttal, including additional clarifications and further experiments, to address your concerns as thoroughly as possible.
>
> We would be very grateful if you could kindly take a moment to review our responses and let us know whether they satisfactorily resolve the issues you raised. If you have any further questions or would like additional clarification, we would be more than happy to provide it.
>
> Thank you once again for your time, consideration, and engagement.
>
> With best regards,
>
> The Authors

---

### Official Review · Reviewer_B5Bb · 2025-11-02

**Soundness:** 2
**Presentation:** 2
**Contribution:** 2
**Rating:** 4
**Confidence:** 3

**Summary:**

The paper proposes LLMBoost, a boosting-inspired ensemble framework that leverages the internal hidden states of large language models rather than treating models as black boxes that only mix inputs or final outputs. It connects multiple models so that later models can directly attend to and fuse the intermediate representations of earlier ones, enabling hierarchical error correction and knowledge transfer. The authors introduce three components: a cross-model attention mechanism for accessing predecessor hidden states; a chain training procedure that progressively fine-tunes each model with an error-suppression objective to correct its predecessor’s mistakes with minimal extra computation; and a near-parallel inference pipeline that passes hidden states layer by layer across models to achieve efficiency close to single-model decoding.

**Strengths:**

1. The paper is well-written and easy to follow.
2. The idea of chain training paradigm is novel, interesting and straightforward.

**Weaknesses:**

1. Given that LLMBoost significantly increases computational cost and GPU memory usage during both training and inference, the performance gains appear relatively modest.
2. LLMBoost introduces a range of hyperparameters, which makes training and inference more complex, and for some of them (e.g., $\beta$ in Equation 6), the paper does not discuss how to set them.
3. The Error-Suppression Objective seems a bit strange to me.  It maximizes the probability margin between the ground-truth token and other tokens, yet a token that differs from the ground truth is not necessarily incorrect. For example, if the ground-truth token is "hello" and the model predicts "hi", aggressively suppressing the probability of "hi" may undermine the model’s ability to generalize.

**Questions:**

1. Given the multi-fold increase in GPU memory during both training and inference with LLMBoost, I believe it should be compared against a larger single model with a similar memory budget. For example, LLMBoost 2×7B vs. Qwen2.5-14B.
2. During backpropagation, does Cross-Model Attention propagate gradients to the preceding model?
3. Given that the models later in the training pipeline correct more error tokens, do they achieve higher accuracy than the earlier models? During inference, why not assign greater weight to the logits of the later models?
4. In Table 1, for configurations like 2×8B or 2×3B, how is VOTE implemented?
5. A closing square bracket is missing in Equation 6.
6. Some of the notation in Figure 2 appears to be incorrect. For example, $L_l^2$ should be $L_l^1$.

---

> ### Author Response · Authors · 2025-11-20
>
> > **Weekness 1： Given that LLMBoost significantly increases computational cost and GPU memory usage during both training and inference, the performance gains appear relatively modest.**
>
> We provide a full and comprehensive discussion of this issue in **Global Response – Concern 1**, and we kindly invite the reviewer to refer to that section for the complete analysis. Here, we briefly summarize the key points: under matched parameter budgets, LLMBoost consistently matches or exceeds the performance of larger single models while achieving lower inference latency through efficient near-parallel execution. These gains stem from complementary multi-model collaboration rather than simple parameter scaling. For detailed results, extended comparisons, and additional justification, please see Concern 1 in our Global Response.
>
> > **Weekness 2： LLMBoost introduces a range of hyperparameters, which makes training and inference more complex, and for some of them (e.g., in Equation 6), the paper does not discuss how to set them.**
>
> In response to the comment that “LLMBoost introduces many hyperparameters (e.g., $\alpha$ and $\beta$ in Equation (6)), and the paper does not fully explain how they are selected,” we clarify the role and choice of these coefficients here. Our loss for the (i)-th model is defined as
>
> $$
> \mathcal{L} _ {i} = \sum_t   \mathcal{L} _ {s,t}    - \alpha\sum_t \log p^{(i)} _ {t}[y_t^{ \ast}]
> $$
>
> where the step-wise auxiliary term is given by
>
> $$
> \mathcal{L}_{s,t}
> = - \log \bigg(
>     \sigma\big(
>         \beta \,(
>             \log p^{(i)}_t(y_t^{*})
>             - \log p^{(i)}_t(\mathrm{ErrToken}^{(i-1)}_t)
>         )
>     \big)
> \bigg)
> \cdot \mathbf{1}\left( \mathrm{ErrToken}^{(i-1)}_t \neq \emptyset \right)
> $$
>
>
> Here, $\alpha$ is a global weight on the token-level log-likelihood, controlling how strongly the (i)-th model is fitted to the ground-truth tokens, while $\beta$ rescales the margin between the probability of the correct token and the error token from the ((i-1))-th model. Together, these two hyperparameters give users flexibility to balance standard supervised learning against the boosting-style correction of previous models’ errors.
>
> To keep the tuning overhead low while still adapting to different tasks, we use a simple two-stage heuristic rather than a full grid search: we first fix $\alpha = 1.0$ and sweep $\beta$ over a small set of candidates, then fix the best $\beta$ and adjust $\alpha$ in a narrow range around 1.0. In practice, we find that a single configuration selected in this way generalizes well across our benchmarks, and the additional tuning cost is negligible compared to model training.
>
> The empirical results are shown below. For the **LLLMboost (Qwen2.5-7B $\times$ 2)** configuration, the model performs best at $\beta = 0.10$:
>
> **Performance comparison of LLLMboost (Qwen2.5-7B $\times$ 2) under different values of $\beta$ (%)**
>
> | Params | $\beta$ | SVAMP | MAWPS | Avg. |
> | --- | --- | --- | --- | --- |
> | $2 \times 7$ B | 0.01 | 87.0 | 92.4 | 89.7 |
> | $2 \times 7$ B | 0.05 | 88.1 | 93.7 | 90.9 |
> | $2 \times 7$ B | 0.10 | 88.3 | 93.7 | **91.0** |
> | $2 \times 7$ B | 0.20 | 89.0 | 92.4 | 90.7 |
> | $2 \times 7$ B | 0.50 | 88.5 | 92.4 | 90.5 |
>
> With  $\beta = 0.10$  fixed, we then sweep over $\alpha$. The results show that the best performance occurs around $\alpha = 0.90$, achieving the highest average accuracy of 91.5%:
>
> **Performance comparison of LLLMboost (Qwen2.5-7B $\times$ 2) under different values of $\alpha$ (%)**
>
> | Params | $\alpha$ | SVAMP | MAWPS | Avg. |
> | --- | --- | --- | --- | --- |
> | $2 \times 7$ B | 1.00 | 87.0 | 93.7 | 90.4 |
> | $2 \times 7$ B | 0.99 | 87.0 | 93.7 | 90.4 |
> | $2 \times 7$ B | 0.95 | 88.2 | 93.7 | 91.1 |
> | $2 \times 7$ B | 0.90 | 88.8 | 94.1 | **91.5** |
> | $2 \times 7$ B | 0.80 | 87.9 | 92.9 | 90.4 |
> | $2 \times 7$ B | 0.50 | 87.9 | 92.4 | 90.2 |
>
> In summary, the two-stage tuning strategy effectively reduces the hyperparameter search space while converging to high-performing configurations. The optimal values $\alpha = 0.90$ and $\beta = 0.10$ achieve the best balance between suppressing harmful gradients and maintaining optimization stability. Therefore, although additional hyperparameters introduce some complexity, our tuning strategy significantly reduces the search overhead and ensures that LLMBoost remains practical, reproducible, and easy to deploy.

---

> > ### Author Response · Authors · 2025-11-20
> >
> > > **Weekness 3**： The Error-Suppression Objective seems a bit strange to me. It maximizes the probability margin between the ground-truth token and other tokens, yet a token that differs from the ground truth is not necessarily incorrect. For example, if the ground-truth token is "hello" and the model predicts "hi", aggressively suppressing the probability of "hi" may undermine the model’s ability to generalize.
> >
> > We appreciate the reviewer’s insightful comment. Indeed, in certain natural language tasks, substitutions such as “hello” and “hi” are semantically similar and differ only in surface expression. Increasing the probability of “hello” while suppressing “hi” could, in principle, introduce a small loss of local diversity. However, such substitutions generally produce only minor perturbations to the model’s probability distribution, which can be easily absorbed and do not induce structural changes in the model’s reasoning trajectory or final output.
> >
> > In contrast, **the Error-Suppression Objective (ESO) plays a crucial corrective role when the predecessor model produces semantically incorrect or logically inconsistent tokens**. For example, in mathematical reasoning, wrong digits, reversed inequality signs, or incorrect logical branches often lead the model toward a completely different reasoning path. The harm caused by such errors is far more severe than the negligible differences between synonymous expressions. Therefore, explicitly suppressing these “genuinely incorrect” tokens provides substantial improvements in reasoning performance.
> >
> > It is important to emphasize that $\mathcal{L}_ {s,t}$ does not operate in isolation. It is a key component of the dual-objective formulation, which balances correction and retention in our boosting-style multi-model collaboration. The standard cross-entropy loss (controlled by $\alpha$) preserves the model’s general linguistic knowledge—for example, that “hello” and “hi” are semantically related—while $\mathcal{L}_ {s,t}$ functions as a conditional expert that activates only when the preceding model $m_ {i-1}$ makes an error, guiding the subsequent model $m_i$ to strongly prefer the ground-truth label $y^{\ast}$ over the incorrect token.

---

> > > ### Author Response · Authors · 2025-11-20
> > >
> > > > **Question 1**: Given the multi-fold increase in GPU memory during both training and inference with LLMBoost, I believe it should be compared against a larger single model with a similar memory budget. For example, LLMBoost 2×7B vs. Qwen2.5-14B.
> > >
> > > Please refer to our Global Response concern 1.
> > >
> > > > **Question 2**: During backpropagation, does Cross-Model Attention propagate gradients to the preceding model?
> > >
> > > No. In our training setup, we use ****unidirectional**** cross-model attention, which ensures that information flows only from the preceding model to the subsequent model, with ****no backward gradient propagation****. The hidden states of the earlier model are treated as auxiliary inputs to the later model, so gradients from the cross-model attention module update only the parameters of the subsequent model and do not flow back to the preceding one. This asymmetric design is intentional: our framework follows a ****boosting-style paradigm****, where the subsequent model is explicitly trained to ****correct the errors and compensate for the weaknesses of the preceding model****. The earlier model produces an initial prediction and intermediate representations, while the later model, via cross-model attention, learns to focus on complementary information that the former fails to capture, thereby preserving model independence and diversity while refining the overall prediction.
> > >
> > > > **Question 3**: Given that the models later in the training pipeline correct more error tokens, do they achieve higher accuracy than the earlier models? During inference, why not assign greater weight to the logits of the later models?
> > >
> > > Our design goal is to encourage the models to form complementary relationships, rather than creating a “weak-to-strong” sequential upgrade where later models are inherently superior. LLMBoost aims to learn a set of expert models that exhibit different preferences and strengths over different regions of the data, enabling the final ensemble to leverage these complementarities for higher overall performance, rather than enforcing a monotonic structure where later models must always be stronger.
> > >
> > > Because of the suppression-based loss, later models are encouraged to maintain lower confidence on samples where earlier models are already confident and correct, and to respond more strongly only when the earlier models are uncertain or wrong. As a result, their probability outputs are optimized for the ensemble as a whole, rather than for a “single-model decision with a larger weight.” Through empirical search, we found that assigning a weight of 0.3 to the subsequent model yields the best performance.
> > >
> > > > **Question 4**: In Table 1, for configurations like 2×8B or 2×3B, how is VOTE implemented?
> > >
> > > We follow the candidate-pool voting strategy introduced in vote[1],  with several adaptations for our scenario. The process is as follows:
> > >
> > > 1. **Independently training multiple models**
> > > We first train multiple homogeneous models on the same dataset, but assign each model a **different random seed and a different data shuffling order**. This yields several model instances with distinct parameter configurations (e.g., two 8B models or two 3B models).
> > > 2. **Generating candidates and constructing the candidate pool**
> > > During inference, for a given input, each model generates multiple candidate answers. For example, in the 2×8B configuration, each model produces 4 candidates, resulting in a total of 8 candidates that form the pool $S = {s_1, s_2, \ldots, s_8}.$
> > > 3. **Similarity-based cumulative voting**
> > > For each candidate (s_i) in the pool, we compute its *cumulative similarity* with all other candidates:$V(s_i) = \sum_{j \neq i} \mathrm{sim}(s_i, s_j).$
> > > The candidate with the highest cumulative similarity ($V(s_i)$) is selected as the final VOTE output.
> > >
> > > [1] Li J, Zhang Q, Yu Y, et al. More agents is all you need[J]. arXiv preprint arXiv:2402.05120, 2024.
> > >
> > > > **Question 5**: A closing square bracket is missing in Equation 6.
> > >
> > > We thank the reviewer for the correction. We have examined Equation (6). This typesetting error has been fixed in the latest version of the paper.
> > >
> > > > **Question 6**: Some of the notation in Figure 2 appears to be incorrect.
> > >
> > > We appreciate the reviewer’s feedback. The corresponding annotations have been corrected and unified in the latest version of the paper to ensure consistency with the formulas in the main text.

---

> ### Author Response · Authors · 2025-11-27
>
> Dear Reviewer B5Bb,
>
> We sincerely appreciate the time and effort you have devoted to reviewing our work, and we are truly grateful for your thoughtful comments and suggestions. Over the past 70 hours, we have carefully prepared our rebuttal, including additional clarifications and further experiments, to address your concerns as thoroughly as possible.
>
> We would be very grateful if you could kindly take a moment to review our responses and let us know whether they satisfactorily resolve the issues you raised. If you have any further questions or would like additional clarification, we would be more than happy to provide it.
>
> Thank you once again for your time, consideration, and engagement.
>
> With best regards,
>
> The Authors

---

### Author Response · Authors · 2025-11-20
**Global Response**

# Global Response

We thank all reviewers for their valuable feedback. The reviewers recognized our work for its strengths across multiple aspects:

- Method Strengths:
    - Recognized as novel and interesting.(B5Bb、tKoq、VpSK)
    - Theory:clean MSE-based guarantee. (VpSK)
- Paper Quality:
    - Well-written. (B5Bb、VpSK)
- Implementation & Impact:
    - Easy to follow. (B5Bb)
- Experiments
    - Good performence.(tKoq)
    - Broad tasks, multiple sizes. (VpSK)

Below, we address common concerns and provide additional experimental results.
## Concern 1:

Reviewers focus on the fairness of method comparison. Given that LLMBoost exhibits a multi-fold increase in GPU memory during both the training and inference phases, it is therefore argued that LLMBoost should be compared with larger-scale single models that have a similar memory budget (e.g., the comparison between LLMBoost 2×7B and Qwen2.5-14B).

## Response1:

In response to the reviewer’s concern, we provide direct comparisons with same-family models under matched parameter scales.

**Conclusion.**
Across configurations with comparable total parameter counts, LLMBoost consistently demonstrates three core advantages over single models of equivalent size:

1. **Higher accuracy at the same or even smaller parameter scale.**
    - At the 14B scale, **LLMBoost (Qwen2.5-7B × 2)** outperforms **Qwen2.5-14B** on both commonsense reasoning (**+0.4**) and arithmetic reasoning (**+0.4**).
    - Even under a reduced parameter budget, **LLMBoost (Qwen2.5-3B × 2, total 6B)** matches or surpasses **Qwen2.5-7B** on multiple arithmetic tasks(**+0.5**)—despite using **1B fewer parameters**.
2. **Lower inference latency with scalable efficiency benefits.**
    - For the 14B configuration, LLMBoost reduces per-token latency from **0.098 s → 0.075 s**, yielding a **23% improvement**.
    - A similar trend holds in the smaller setting: **LLMBoost (Qwen2.5-3B × 2)** achieves **0.043 s/token**, which is **23% faster** than the **7B** model (0.056 s/token), even though the total parameter count is slightly lower (6B vs. 7B).
    - These results show that multi-model parallelism introduces minimal overhead and can *even outperform* single larger models in runtime.
3. **Bridging the deployment gap between small and very large models.**
    - In real-world deployment, **Llama3.1-70B models or other large models are often infeasible** due to memory and latency constraints, while relying solely on **smaller Llama3.1-8B models underutilizes GPU capacity** and significantly limits performance.
    - Although a 20B–60B model could in principle be obtained through full pretraining, the required computational cost is often impractical. Moreover, the **Llama-3 family does not include any models in this intermediate range**, leaving a noticeable gap between the 8B and 70B variants. Our approach offers a practical alternative: by boosting several moderate-size models (e.g., Llama3.1-8B $\times$ 3), we approximate a “virtual 24B model” using only **lightweight fine-tuning**, avoiding the need to pretrain a new mid-scale model while still achieving competitive performance.
    - Empirically, these boosted ensembles achieve **performance approaching that of 70B-scale models**, while maintaining **substantially lower deployment cost**, **better scalability**, and **flexible resource utilization**. Moreover, thanks to parallel decoding across moderate-size models, the ensemble often delivers **faster inference** than a monolithic 70B model. This effectively closes the otherwise missing middle ground between small models and extremely large models.

Overall, the improvements come from **effective multi-model collaboration and complementary error correction**, rather than merely increasing parameter count.

---

> ### Author Response · Authors · 2025-11-20
> **Concern 1（2/3）**
>
> The detailed results are presented in the tables and analysis below.
>
> ### Results on Arithmetic Datasets:
>
> | Model | Params | Latency | AQuA | GSM8k | MAWPS | SVAMP | Avg |
> | --- | --- | --- | --- | --- | --- | --- | --- |
> | Qwen2.5-14B | 14B | 0.098 | 62.2 | 78.5 | 91.6 | 87.0 | 79.8 |
> | LLMBoost (Qwen2.5-7B × 2) | 14B | **0.075** | 59.8 | 78.1 | 94.1 | 88.8 | **80.2** |
> | Qwen2.5-7B | 7B | 0.056 | 56.8 | 75.1 | 92.0 | 86.1 | 77.5 |
> | LLMBoost (Qwen2.5-3B × 2) | **6B(-1B)** | **0.043** | 60.2 | 74.0 | 92.4 | 85.3 | **78.0** |
>
> The above results indicate that when the total parameter is the same, LLMBoost (3B $\times$ 2) has an average score of 78.0 across the five tasks, which is better than the standalone Qwen2.5-7B (77.5); similarly, Qwen2.5-7B $\times$ 2 is also better than Qwen2.5-14B.
>
> ### Results on Commonsense Datasets:
>
> | Model | Params | Latency | PIQA | WinoG | HellaS. | Boolq | SIQA | OBQA | Avg |
> | --- | --- | --- | --- | --- | --- | --- | --- | --- | --- |
> | Qwen2.5-14B | 14B | 0.098 | 90.2 | 85.8 | 92.9 | 71.2 | 78.9 | 91.4 | 85.1 |
> | LLMBoost (Qwen2.5-7B × 2) | 14B | **0.075** | 89.5 | 86.7 | 95.1 | 71.6 | 79.1 | 91.6 | **85.6** |
> | Qwen2.5-7B | 7B | 0.056 | 86.7 | 83.6 | 89.4 | 68.7 | 77.6 | 87.4 | 82.3 |
> | LLMBoost (Qwen2.5-3B × 2) | **6B(-1B)** | **0.043** | 85.9 | 81.8 | 91.4 | 69.7 | 79.0 | 86.2 | **82.3** |
>
> Across the six commonsense reasoning benchmarks, LLMBoost (Qwen2.5-7B $\times$ 2, 14B) achieves an average accuracy of 85.6, which outperforms the larger single model Qwen2.5-14B (85.1), while substantially outperforming the single 7B model (82.3) with notable gains such as +2.8 on PIQA, +3.1 on WinoG, and +5.7 on HellaSwag. Smaller-scale configurations exhibit the same pattern: LLMBoost (Qwen2.5-3B $\times$ 2, 6B) reaches 82.3, matching the single 7B (82.3) despite using 1B fewer parameters.
>
> These explicit results demonstrate that LLMBoost’s gains arise not from parameter scaling, but from effective multi-model collaboration and complementary error correction, enabling smaller ensembles to consistently match or outperform larger single models under comparable or even lower memory budgets.
>
> ### End-to-End Inference Time Comparison of Different Models
>
> | Model | End-to-End Wall Time (s) | Token Cost (s) |
> | --- | --- | --- |
> | Qwen 14B | 17.40 | 0.098 |
> | LLMBoost (Qwen2.5-7B $\times$ 2, parallel, GPU=2) | 13.75 | **0.075** |
> | LLMBoost (Qwen2.5-7B $\times$ 2, parallel, GPU=1) | 14.87 | 0.081 |
>
> Beyond accuracy, LLMBoost also brings clear efficiency advantages. Thanks to our near-parallel decoding framework, two Qwen2.5-7B models can run concurrently, reducing single-token latency from 98 ms (Qwen2.5-14B) to 75 ms (LLMBoost Qwen2.5-7B × 2),a 23% speedup while maintaining comparable or slightly better performance. This advantage is also reflected at the end-to-end level: LLMBoost (Qwen2.5-7B  $\times$ 2, parallel, GPU=2) achieves an inference wall time of 13.75 s, substantially outperforming other baselines.
>
> Even in resource-constrained environments where all models must run on a single GPU, LLMBoost remains competitive: the 7B  $\times$ 2 single-GPU configuration reaches 14.87 s, still faster than larger single models.

---

> > ### Author Response · Authors · 2025-11-20
> > **Concer 1 （3/3）**
> >
> > ### LLMBoost (Llama3.1-8B × 3) bridges the performance gap between Llama3.1-8B and Llama3.1-70B.
> >
> > | Model | Params | Latency | PIQA | WinoG | HellaS. | Boolq | SIQA | OBQA | Avg |
> > | --- | --- | --- | --- | --- | --- | --- | --- | --- | --- |
> > | Llama3.1-70B | 70B | 0.254 | 93.2 | 93.1 | 96.9 | 76.5 | 82.4 | 92.0 | 89.2 |
> > | LLMBoost (Llama3.1-8B × 3) | 24B | 0.064 | 87.4 | 86.3 | 95.2 | 71.7 | 80.7 | 84.8 | 84.5 |
> > | LLMBoost (Llama3.1-8B × 1) | 8B | 0.038 | 83.3 | 81.8 | 90.6 | 69.5 | 75.8 | 79.0 | 80.0 |
> >
> > LLMBoost provides a practical option in settings where running a 70B model is challenging, while a single 8B model may be insufficient. The results show that LLMBoost achieves competitive task-level performance relative to the 70B model. For example, on **HellaSwag**, LLMBoost (Llama3.1-8B $\times$ 3) reaches **95.2**, only **1.7 points** below Llama3.1-70B, and on **SIQA**, the gap is similarly small (**80.7 vs. 82.4**). These observations suggest that LLMBoost (Llama3.1-8B $\times$ 3) can approach the reasoning capability of a much larger 70B model on several representative benchmarks.
> >
> > At the overall level, LLMBoost (Llama3.1-8B $\times$ 3) achieves an average accuracy of **84.5**, positioning its performance between Llama-8B and Llama-70B. This indicates that LLMBoost can naturally **fill the gap** between small models and very large models by combining several moderate-size models.
> >
> > Importantly, this boosted capability is obtained **using only lightweight fine-tuning**, without the need to pretrain a new mid-scale model from scratch. The ensemble also offers **lower inference latency** (e.g., **0.064 s/token** vs. **0.254 s/token** for Llama-70B), making it a practical option in settings where running a 70B model is challenging while a single 8B model may be insufficient.
> >
> > Overall, these results indicate that LLMBoost delivers not only accuracy improvements but also significantly lower latency and faster end-to-end runtime compared to both larger single models and traditional ensemble methods, underscoring its practical value under real-world deployment constraints.

---

> > > ### Author Response · Authors · 2025-11-20
> > > **Concer 2**
> > >
> > > ## Concern2:
> > >
> > > Both reviewers highlight a central limitation of the current method: LLMBoost primarily operates within a single model family (e.g., LLaMA-3), and its ability to support truly heterogeneous model ensembles remains unclear.
> > >
> > > ## Response2:
> > >
> > > We thank the reviewer for the insightful suggestion regarding *heterogeneous model ensembles*. We indeed explored incorporating models of different scales and architectures in the early stages of development, but encountered two structural challenges that fundamentally affect LLMBoost’s core mechanism of **cross-model state awareness**. We plan to address these challenges in future work:
> > >
> > > (1) **Tokenizer inconsistency.** Different model families segment text into incompatible token representations, preventing the output of one model from serving as valid input to the next.
> > >
> > > (2) **Hidden-size mismatch.** The hidden-state dimensionalities of different architectures do not align, making it impossible for the subsequent model to directly consume the previous model’s representations.
> > >
> > > Because of these limitations, fully cross-architecture heterogeneous ensembles are currently **structurally infeasible**. One potential direction is to learn a shared high-dimensional latent space that allows hidden states from different model families to be projected into a compatible representation. However, building such a space requires nontrivial architectural and optimization efforts, including the design of stable projection operators and the maintenance of representational consistency across models. These challenges go beyond the scope of the current work and are therefore better left for future research.
> > >
> > > To respond to the reviewer’s suggestion as much as possible, we conducted a *weakly heterogeneous* experiment using **LLaMA3.1-8B** and **LLaMA3.2-3B**—a pair of models with compatible tokenizers, similar architectures, but different scales. To enable interaction between these models, we introduced two lightweight and practical adaptation strategies:
> > >
> > > - **Layer chunking:** Construct block-level correspondences between deeper and shallower models;
> > > - **Learnable projection:** Map the previous model’s hidden states into the dimensionality of the next model.
> > >
> > > Based on these techniques, we successfully implemented an 8B+3B weakly heterogeneous version of LLMBoost and observed **consistent improvements** across multiple benchmarks:
> > >
> > > **Arithmetic reasoning**
> > >
> > > | Model | AQuA | GSM8k | MAWPS | SVAMP | Avg |
> > > | --- | --- | --- | --- | --- | --- |
> > > | Llama3.2-3B | 31.9 | 49.7 | 84.9 | 65.8 | 58.1 |
> > > | Llama3.1-8B | 42.3 | 63.7 | 89.5 | 77.4 | 68.2 |
> > > | **LLMBoost (8B+3B)** | **43.7** | **64.4** | **89.7** | **78.6** | **69.1** |
> > >
> > > **Commonsense reasoning**
> > >
> > > | Model | PIQA | WinoG | HellaS. | BoolQ | SIQA | OBQA | Avg |
> > > | --- | --- | --- | --- | --- | --- | --- | --- |
> > > | Llama3.2-3B | 78.5 | 77.7 | 83.9 | 62.4 | 74.3 | 76.8 | 75.6 |
> > > | Llama3.1-8B | 83.3 | 81.8 | 90.6 | 69.5 | 75.8 | 79.0 | 80.0 |
> > > | **LLMBoost (8B+3B)** | **85.9** | **85.0** | **93.8** | **71.3** | **80.4** | **84.0** | **83.7** |
> > > |  |  |  |  |  |  |  |  |
> > >
> > > LLMBoost already supports *near-heterogeneous* configurations and provides a promising foundation for future extensions toward broader cross-family and domain-specialized model combinations.

---

### Author Response · Authors · 2025-11-27

Dear Reviewers B5Bb, tKoq, and VpSK,

We sincerely appreciate the time and effort you have dedicated to reviewing our work, and we are very grateful for your valuable suggestions. We have spent over 70 hours preparing this rebuttal, including additional clarifications and new experiments. We genuinely hope that our responses adequately address your concerns, and we would be grateful if you could kindly take a moment to review them and let us know whether any issues remain.

If you have any further questions or comments, we would be more than happy to address them.

Thank you very much for your consideration and engagement.

Best regards,

The Authors

---

### Meta-Review · Area_Chair_i3sX · 2026-01-07

**Summary:**

The reviewers' main concerns can be summarized as follows:
- **Limited applicability to heterogeneous models.** The framework seems to require each LLM to take the identical (or similar) form, which greatly limits the possibility of achieving improved performance by taking advantage of having many models with different strengths.
- **Lack of comparison with equi-capacity models.** The framework does not make a formal comparison with unibody LLMs with similar computational footprint.
- **Efficiency comparisons.** The paper requires additional discussions on the computational footprint (both training and inference) w.r.t. other frameworks.
- **Long-context reasoning.** A reviewer has suggested evaluation on benchmarks that require a longer reasoning.

Some of the concerns have been addressed well, but the limited applicability has not been resolved, and is indeed a critical weakness of the proposed method. In addition, the AC does have some concerns regarding the evaluation w.r.t. the larger-scale unibody LLMs. In particular, (1) the reported results for the models seems to diverge significantly from those of the official technical report (are they instruction-tuned?); (2) are these models also tuned similarly, using the comparable computational footprint to the proposed framework?

Thus I recommend rejection.

**Reviewer Concerns:**

- **Limited applicability to heterogeneous models.** This was partially resolved, but it is unlikely that the proposed method can be extended easily to different model architectures.
- **Lack of comparison with equi-capacity models.** This is resolved, but as noted above, there are some concerns regarding the provided results.
- **Efficiency comparisons.** This has been resolved.
- **Long-context reasoning.** This has been resolved.

**Reviewer Scores:**

It is likely that B5Bb might have considered raising the score. However, multiple concerns of the other two reviewers seems to remain less resolved.

---

### Decision · Program_Chairs · 2026-01-26

Reject